

# How well do hydrological models learn from limited discharge data? A comparison of process- and data-driven models

Maria Staudinger[1], Anna Herzog[2], Ralf Loritz[3], Tobias Houska[4], Sandra Pool[5], Diana Spieler[6,7], Paul D. Wagner[8], Juliane Mai[9], Jens Kiesel[8,12], Stephan Thober[10], Björn Guse[8,11], and Uwe Ehret[3]

[1]Department of Geography, University of Zurich, Winterthurerstrasse 190, 8057 Zurich, Switzerland

[2]Department of Hydrology and Climatology, Institute of Environmental Science and Geography, University of Potsdam, Potsdam, Germany

[3]Institute of Water and Environment, Karlsruhe Institute of Technology (KIT), Karlsruhe, Germany

[4]Department of Landscape Ecology and Resources Management, University of Gießen, Gießen, Germany

[5]Department Water Resources and Drinking Water, Eawag - Swiss Federal Institute of Aquatic Science and Technology, Dübendorf, Switzerland

[6]Department of Hydrosciences, Institute of Hydrology and Meteorology, TUD Dresden University of Technology, Dresden, Germany

[7]now at: Schulich School of Engineering, University of Calgary, Calgary, Canada

[8]Department of Hydrology and Water Resources Management, Institute for Natural Resource Conservation, Kiel University, Kiel, Germany

[9]Earth and Environmental Science, University of Waterloo, Waterloo, Ontario, Canada

[10]Computational Hydrosystems, Helmholtz Center for Environmental Research - UFZ, Leipzig, Germany

[11]German Research Centre for Geosciences, Section Hydrology, Potsdam, Germany

[12]Stone Environmental, 535 Stone Cutters Way, 05602 Montpelier (VT), USA

**Correspondence:** Maria Staudinger (maria.staudinger@geo.uzh.ch)

**Abstract.** A widespread assumption is that data-driven models only achieve good results with sufficiently large training data, while process-based models are usually expected to be superior in data-poor situations. In our study, we investigate this assumption by calibrating several process-based and data-driven hydrological models with training data sets of observed discharge that differ in the number of data points and the type of data selection. The tested models include four commonly used process-based

models (GR4J, HBV, mHM, and SWAT+) and four data-driven models (conditional probability distributions, regression trees, ANN, and LSTM), which are calibrated for three meso-scale catchments representing three different landscapes in Germany: the Iller in the Alpine region, the Saale in the low mountain ranges, and the Selke in the Central German lowlands. We used conditional entropy to evaluate model performance and the learning capability of a model (i.e., change in model performance with increasing sample size).

In addition to the main question of this study, i.e., to what extent the performance of the different models depends on the training data set, we also investigated whether the selection of the training data (random or according to information content, selection of contiguous time periods, or independent time points) plays a role. We also investigated whether there is a relationship between the information contained in the data and the shape of the learning curve for different models that allows prediction of the achievable model performance, and whether the use of more spatially distributed model inputs leads

to improved model performance compared to spatially lumped inputs.



Process-based models outperformed data-driven models for small amounts of training data due to their predefined structure based on process representation. However, with increasing amounts of training data, the learning curve of process-based models quickly saturates, and using about 2 to 5 years of training data, the data-driven LSTM consistently outperforms all process-based models. In particular, the LSTM continues to learn from more training data without approaching saturation. Sur-
prisingly, fully random sampling of training data points for the HBV model leads to better learning results not only compared to consecutive random sampling but also compared to optimal sampling in terms of information content. Analyzing multivariate catchment data allows predictions about how these data can be used to predict discharge. When no memory was considered, the conditional entropy was large, but as soon as some memory was introduced in the form of a past day or past week, the conditional entropy became smaller, suggesting that memory is a very important component in the data and that capturing it
improves model performance. This was particularly the case for the catchment from the low mountain ranges and the Alpine region.

## 1 Introduction

Hydrological predictions are often made using process-based models whose predefined structure (Devia et al., 2015), variables, and parameters reflect – in a simplified way – our understanding of how a catchment partitions, stores and releases water. In
contrast, data-driven models have a statistical background and are built specifically for a catchment or a region using only available data. Recently, data-driven models have been shown to perform equally well or better than established process-based models in different applications such as rainfall-runoff modelling (Kratzert et al., 2018; Mai et al., 2022; Girihagama et al., 2022; Xiang et al., 2020), flood forecasting (Zhang et al., 2022), or groundwater level forecasting (Mohanty et al., 2015; Daliakopoulos et al., 2005). A common assumption in the hydrological community is that data-driven models perform well with
sufficiently large training data sets, while process-based models are superior in data-poor situations. As opposed to process-based models, data driven models, especially Long Short-Term Memory networks (LSTMs), generally perform better and are more robust when trained on large sample datasets covering hundreds of catchments with long time series (Kratzert et al., 2024). This is not surprising given that LSTMs are general-purpose architectures with no built-in hydrological knowledge, such as conservation of mass, and not specifically designed for rainfall-runoff modelling. As such, they must learn the relationship
between meteorological variables and the discharge from the data itself each time they are trained, since their weights are randomly initialized before the training. Consequently, test results for catchments improve when LSTMs are trained regionally (e.g. Loritz et al., 2024). In contrast, process-based hydrological models are developed specifically to represent the hydrological system and embed prior knowledge of hydrological processes. Some process-based models have been developed to allow for validation in space, and in this type of process-based models, the representation of hydrologic fluxes at different resolutions
is considered (Rakovec et al., 2016). This leads us to the following research question: How well do different models learn if provided with limited discharge data, and more specifically is there a break point between process-based and data-driven models (Q1)?





What constitutes a sufficiently large training set is not straightforward to define. For process-based models, it is generally recommended to use long continuous discharge records for model training/calibration (Vrugt et al., 2006; Yapo et al., 1996; Shen et al., 2022; Mai, 2023). The idea behind this recommendation is that long records contain information on processes occurring under a range of hydrological conditions (e.g., low, mean, and high flows, or extremes) and at different temporal scales (e.g., event, season, years). However, many regions lack such records, and it is therefore important to understand how much data are needed to obtain a model with satisfactory discharge simulations. Work with process-based models and catchments with contrasting climate has shown that much of the hydrological information relevant for model training is theoretically represented in a few data points (Wright et al., 2018) covering less than 10% of a longer time period (Singh and Bárdossy, 2012; Perrin et al., 2007). In practice, this means that a continuous time series of a few months may already be informative enough to achieve a model performance similar to that when using a time series of a year or more (Brath et al., 2004; Melsen et al., 2014; Sun et al., 2017). For example, results from Seibert and Beven (2009) and Pool and Seibert (2021) suggest that about twelve to sixteen discharge observations during peak flows or events and their subsequent recessions can contain much of the information of longer continuous time series. Several authors have examined the characteristics of the most valuable subsets of a longer time series. They have typically emphasized the importance of having a sample that represents the natural variability of flow and covers the wetter, and hydrologically active periods (Harlin, 1991; Singh and Bárdossy, 2012; Sun et al., 2017; Vrugt et al., 2006; Yapo et al., 1996; Zhang et al., 2023). It may also be worthwhile to collect discharge data in a previously ungauged catchment (Correa et al., 2016; Rojas-Serna et al., 2006; Pool et al., 2017; Zhang et al., 2023). Previous research has shown that limited data availability significantly affects the performance of data-driven models. Acuña Espinoza et al. (2024) found that training an LSTM on a small, non-diverse dataset can limit not only its test performance, but also its ability to extrapolate to unseen hydrological states. The results of Snieder and Khan (2025) suggest that diverse training data are more valuable, allowing sub-setting of repetitive datasets using diversity-based sampling.

These studies encourage the use and strategic collection of short discharge records to calibrate process-based models, but it remains to be tested how well data-driven models perform in a data-scarce context. And it remains to be tested how random sampling, optimizing information content, or providing continuous or independent time points affects the learning of models. We therefore address the following additional research question: How does the scheme of selecting training data affect model performance (here: rainfall runoff modeling) (Q2)?

Similarly, all datasets that are used in catchment hydrological modelling contain data that may be either informative, redundant, or even dis-informative. It would be advantageous to be able to derive from a prior data analysis both a) the optimal model type and b) the minimum training data requirements for a given catchment and the data sets provided. Such an analysis would reduce the overall time and effort required. So we ask: does analyzing the information content of catchment data allow predictions about the performance of different model types (Q3)? As a special but typical case of the ability of models to exploit information contained in data, we further ask whether spatially distributed meteorological forcing data contain relevant information and thus enhance learning without compromising the generality of what has been learned (Q4).

The remainder of this paper is structured as follows: In Section 2, we present the catchments, data sets, hydrological models and performance measures used in the study. In the same section we also describe four experiments E1 - E4, which were



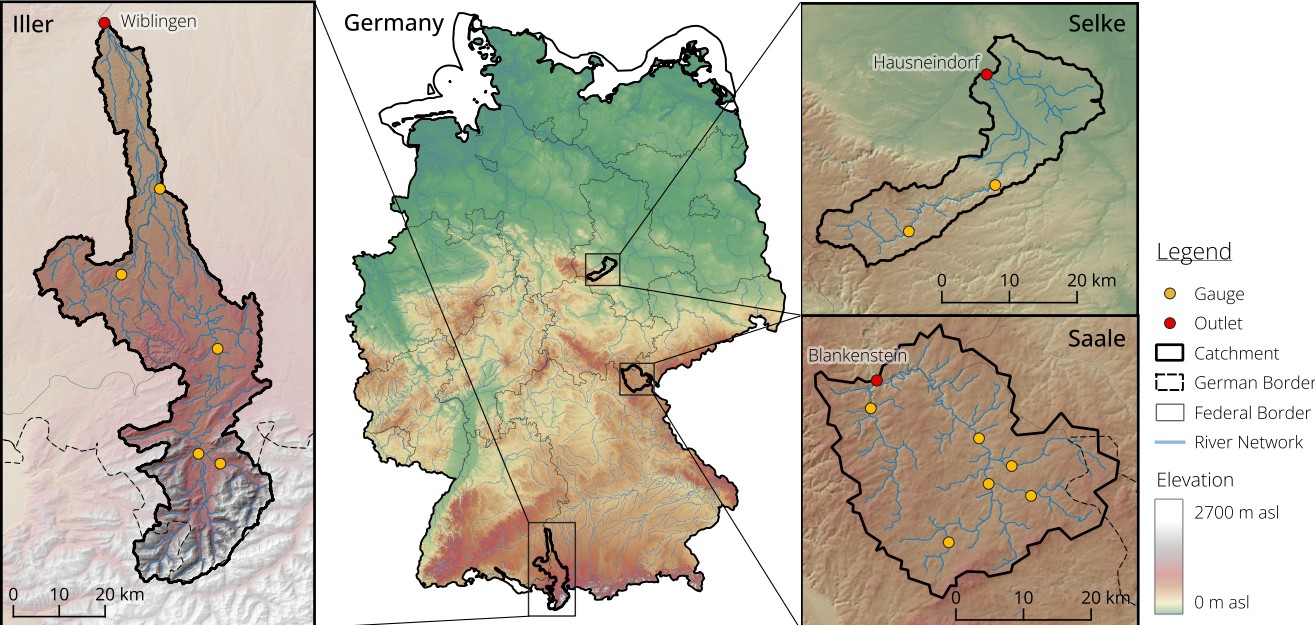

**Figure 1.** Geographic location, topography and gauging location of the sub-basins of the study catchments.

designed to address the questions Q1 - Q4. In Section 3 we present and discuss the results of E1 - E4. There we also discuss the limitations of our study and the advantages of using information measures for system analysis and model performance evaluation. Finally, in Section 4, we draw conclusions and point to future research.

## 2 Methods and data

### 2.1 Study areas

We selected three meso-scale catchments representing three different hydrologic regions of Germany: the Iller in the Alpine region, the Saale head water in the German low mountain range, and the Selke in the central German lowlands. This choice was made because we expect different processes to be more important in each of these catchments if we model them appropriately. For example, snow-related processes should be most important for the Iller catchment. Having these three example catchments allows to have a closer look at the processes that can explain model performance and the learning capabilities of specific models focusing less on spatial diversity but more on investigating the information content within time series. The location of the catchments within Germany and their topography are shown in Figure 1 while Table 1 provides some summary information for each catchment.





**Table 1.** Overview of catchment characteristics for the three study catchments, Iller, Saale and Selke.

|  | Iller at Wiblingen | Saale at Blankenstein | Selke at Hausneindorf |
|---|---|---|---|
| Size [km$^2$] | 2140 | 1011 | 461 |
| Mean elevation [m asl] | 906 | 576 | 262 |
| Elevation range [m asl] | 475 - 2584 | 412 - 851 | 105 - 590 |
| Regime | nival | nivo-pluvial | pluvial |
| Mean annual P [mm] | 1500 | 836 | 660 |
| Mean annual Q [mm] | 745 | 367 | 102 |

### 2.1.1 Iller

The Iller catchment area up to gauge Wiblingen is $2140\,\mathrm{km}^2$ and has a diverse topography, including mountainous regions in the south with elevations above $2000\,\mathrm{m}$ asl and lower, flatter areas in the North. Approximately 50% of the catchment area is cropland and pastures, about 30% is covered by forests, predominantly mixed and coniferous forests, about 10% is urban areas, and about 10% is covered by bare rock, sparsely vegetated areas, peat and water bodies. A variety of soil types are found in the catchment, including lithosols and cambisols (shallow, rocky, and well-drained), which predominate in the southern mountainous regions and cover about 25% of the catchment. Cambisols (well-drained) occupy about 30% of the catchment area and are found primarily in the mid-elevation regions. Alluvial soils (well-drained) are prevalent in the northern part of the catchment, along river valleys, and comprise about 20% of the total area. Gleysols (poorly drained and often waterlogged) are found in wetter, low-lying areas and wetlands and comprise about 10% of the catchment. The bedrock geology consists of Mesozoic limestone and dolomite in the Alpine region. North of the Alps, the bedrock transitions to the Molasse Basin, which consists of Tertiary sedimentary rocks, including sandstones, marls, and conglomerates. The northern plains are dominated by Quaternary alluvial deposits of gravel, sand, silt, and clay.

### 2.1.2 Saale

The catchment area of the Saale at gauge Blankenstein is $1011\,\mathrm{km}^2$ with a varied topography, where the upper regions are hilly to mountainous (elevations around 700-900 m asl) and the downstream areas have more gentle slopes. About 60% of the catchment area is used for agriculture and pasture, about 30% of the catchment area is covered by forests, mainly coniferous, and about 10% are urban areas (see also Guse et al. (2019)). Podzols (well-drained) are prevalent in forested areas and cover about 20% of the catchment. Cambisols (well-drained) cover about 30% of the catchment and are found in both agricultural and forested areas. Gleysols (poorly drained and often waterlogged) are found in wetter, low-lying areas and wetlands and cover about 10% of the catchment. Loess soils (highly productive) and alluvial soils (well-drained) each cover about 20% of the catchment. In the upper part of the catchment, the geology consists of metamorphic and igneous rocks (schists, gneisses and granites). To the north, the geology changes to Triassic sedimentary rocks, including sandstones, marls, and limestones. Along the river valleys and floodplains, Quaternary alluvial deposits of gravel, sand, silt, and clay dominate.



### 2.1.3 Selke

120

The Selke catchment at gauge Hausneindorf covers $463\,\mathrm{km}^2$. The catchment has varied topography, with the upper regions being mountainous and more gentle slopes in the downstream areas. Approximately 55% of the catchment area is used for agriculture, about 35% of the catchment is covered by forests, predominantly mixed and coniferous forests. Around 7% urban area and about 3% natural grasslands, wetlands, and water bodies. Podzols (well-drained) are prevalent in forested areas,

125 covering about 25% of the catchment. Cambisols (well-drained) cover around 30% of the catchment and are found in both agricultural and forested areas. Gleysols (poor drainage and often waterlogged) are present in wetter, low-lying areas and wetlands, covering about 10% of the catchment. Loess Soils (highly productive) cover about 15% of the catchment. Alluvial Soils (well-drained) are found along river valleys, covering about 20% of the area. Schist and clay stone are found in the mountain area, tertiary sediments with loess soil in the lowland areas. Highly permeable Quaternary alluvial deposits dominate

130 along river valleys and floodplains.

### 2.2 Data

For each catchment, static catchment properties and dynamic data were collected. The static data comprise information about soils (horizon depth, sand and clay content), land use (classes from the CORINE map (CLMS, 2019) for SWAT+ and reclassified to forest, urban and pervious for mHM), and topography. The dynamic data comprise precipitation (P), air temperature (T)

135 and estimated potential evapotranspiration (PET) in form of gridded data as well as discharge (Q) measured at the outlet of the catchment and is available for the period 2000 - 2015. The available data, their temporal and spatial resolution are summarized in Table 2.

**Table 2.** Static and dynamic input data for each catchment. The dynamic input data (time series of precipitation, potential evapotranspiration, air temperature and discharge) is available for the period 2000-2015.

| Variable | Data/Map/Method | Resolution | Source |
|---|---|---|---|
| Digital Elevation Model | DEM100 | $100\,\mathrm{m}$ | Yamazaki et al. (2019) |
| Land Cover | CORINE | $100\,\mathrm{m}$ | CLMS (2019) |
| Soil map | Soil map (BÜK200) | $100\,\mathrm{m}$ (resampled) | BGR |
| Precipitation | Station data, interpolated | daily, $1\,\mathrm{km}$ | DWD |
| Air temperature | Station data, interpolated | daily, $1\,\mathrm{km}$ | DWD |
| Potential evapotranspiration | Hargreaves and Samani (1985) | daily, $1\,\mathrm{km}$ | based on DWD variables |
| Discharge | Gauge observations | daily | Local authorities |

Gridded values for temperature and precipitation were obtained by interpolating the observations from meteorological stations. PET was estimated using the Hargreaves and Samani equation (Hargreaves and Samani, 1985), with minimum and

140 maximum daily air temperatures from the meteorological stations and subsequent spatial interpolation. Details of the method





can be found in Boeing et al. (2022). As some process-based models require station-based or lumped input data, weighted averages of the grid cell values contributing to individual sub-basins or HRUs were generated.

## 2.3 Hydrological Models

To address our research questions, we set up and applied four process-based and four data-driven hydrological models.

### 2.3.1 Process-based lumped and semi-distributed models

The process-based models GR4J, HBV-light, mHM, and SWAT+ are all provided with the same meteorological data (precipitation, temperature, potential evapotranspiration), linked with, and run through the Python framework SPOTPY (Houska et al., 2015). SPOTPY allows for greater consistency between the runs of the different models, ensuring that all the sampling and input data are exactly the same.

**GR4J** GR4J (Génie Rural à 4 paramètres Journalier) is a daily lumped rainfall-runoff model designed for hydrological simulation and streamflow forecasting (Perrin et al., 2003). The model is parsimonious, requiring only four parameters, and has provided reliable results with minimal data requirements in the past (Smith et al., 2019; Kuana et al., 2024). The four parameters that are used in the standard model variant are: Maximum capacity of the production store (mm), groundwater exchange coefficient (mm/d), one-day-ahead maximum capacity of the routing store (mm) and time base of the unit hydrograph (d). The production store represents soil moisture processes, including infiltration and evaporation. The percolation and baseflow routine simulates groundwater contributions. The routing store manages the flow routing through the catchment. The flood routing itself is done using a unit hydrograph that accounts for the temporal distribution of the runoff. To improve discharge modelling in catchments influenced by snow, GR4J is often combined with the CemaNeige snow module (Valéry et al., 2014) that comes with two additional parameters. In this paper, we use the GR4J and CemaNeige implementations that are provided through the Raven hydrological modelling framework as they are perceived as exact emulations of the original models (Craig et al., 2020). The Raven GR4J implementation offers the possibility to run GR4J in a semi-distributed fashion. We used a subbasin delineation for better input data representation. With only six parameters this model is expected to perform well also under parsimonious calibration strategies.

**HBV** The HBV model is a semi-distributed model, i.e. a catchment can be divided into different elevation and vegetation zones as well as into different sub-basins. The model consists of several model routines and simulates catchment discharge based on time series of precipitation and air temperature as well as estimates of potential evaporation rates. We used it in the version HBV light (Seibert and Vis, 2012) and divided the catchment only in elevation zones as well as sub-basins not explicitly accounting for different land cover.

In the snow routine, snow accumulation and snowmelt are calculated using a degree-day method. Meltwater and precipitation are retained in the snow pack until they exceed a certain fraction of the water equivalent of the snow. Liquid water in the snow pack refreezes according to a refreezing coefficient. The soil routine simulates groundwater recharge and actual evaporation as a function of actual water storage. Actual evaporation from the soil box is either the potential evaporation or linearly reduced with decreasing soil moisture. In the response routine, discharge is calculated as a function of water storage. Groundwater





recharge is added to the upper groundwater box and percolates from there to the lower groundwater box. Finally, a triangular
weighting function is applied in the routing routine to simulate the routing of the runoff to the catchment outlet. When different
elevation zones are used in the model, changes in precipitation and temperature with elevation are taken into account. HBV
has a relatively small total number of model parameters, allowing the use of parsimonious calibration strategies. In our set-up
we used 11 parameters to be calibrated plus 4 parameters that were fixed to default values.

**mHM** The mHM model (Kumar et al., 2013; Samaniego et al., 2010) is a spatially distributed model that accounts for the
major processes of snow accumulation and melt, evapotranspiration, canopy interception, soil water infiltration and storage,
percolation, and runoff generation. These processes are conceptualized as water fluxes between internal model states. Snow
accumulation and melting are estimated using a degree-day method. A multi-layer discretization represents the soil moisture
dynamics in the root-zone, where the lowest layer is spatially variable in depth depending on the soil map.

Evapotranspiration from the soil layers is estimated as a fraction of the potential evapotranspiration depending on the soil
moisture stress and the fraction of vegetation roots present in each layer. Runoff generation is formalized as the sum of direct
runoff, slow and fast interflow, and base flow components. The runoff generated at each grid cell is routed to the outlet using the
Muskingum-Cunge algorithm (Thober et al., 2019). Unique to mHM is the way effective model parameters are estimated using
multi-scale parameter regionalization (MPR) (Kumar et al., 2013; Samaniego et al., 2010). Parameters (e.g., soil porosity)
are estimated based on physiographic properties (e.g., sand and clay content) and transfer functions (e.g., pedo-transfer func-
tions). These transfer functions depend on transfer or global parameters (e.g., factors of the pedo-transfer functions) that are
time-invariant and location-independent. Because of the MPR, the mHM model is more constrained than other process-based
models. Therefore, mHM often provides good results without calibration, using the default parameter set.

**SWAT+** The Soil Water Assessment Tool Plus (SWAT+) is a continuous, semi-distributed eco-hydrological model. It is
a restructured version of the original SWAT (Arnold et al., 1998; Bieger et al., 2017), designed to simulate the effects of
land management and climate on hydrological processes and water quality. The catchment is divided into sub-basins that are
further subdivided into Hydrologic Response Units (HRUs) that each represent a unique combination of land use, soil type, and
topographic conditions within a sub-basin. Soil water content is continuously updated based on the balance of incoming water
(precipitation and irrigation) and outgoing water (evapotranspiration, runoff, lateral flow, and percolation) for each HRU. The
Curve Number method is used to divide precipitation into surface runoff and infiltration. Actual evapotranspiration is calculated
based on water storage in soil, plant characteristics, and open water bodies. Percolation is simulated by tracking the movement
of water from the root zone to deeper soil layers, and eventually to groundwater. Percolation rates depend on soil properties,
soil moisture levels, and the amount of water available after accounting for evapotranspiration and surface runoff. Groundwater
flow is routed through user-defined aquifers and contributes to discharge based on storage and retention parameters. SWAT+
allows the calibration of a large number of parameters, leading to considerable model flexibility, but therefore usually requires
less parsimonious calibration strategies and a higher degree of user knowledge.

An overview of the different resolutions and aggregations of the input data for the process- based models is given in Table 5.





### 2.3.2 Data-driven models

We selected four data-driven models with the aim of covering a wide range of model complexity, from simple (EDDIS and RTREE) to more sophisticated (ANN and LSTM). Details of each model are described below.

**Empirical discrete distributions (EDDIS)** This approach represents the case where there is no prior knowledge of the structure of the real-world system, and therefore the model can only learn from the available training data. The model is deliberately kept as simple as possible to serve as a lower benchmark, and consists of the multivariate joint discrete (binned) distribution of all available training data, including the desired model output (here: discharge). The binning method is explained in Sect. 2.4. As the model is built directly from the training data, no training is required. Applying the model consists of binning

a given set of input data, and then retrieving the conditional discrete distribution of the output given the input from the joint distribution. If necessary, this probabilistic prediction can be reduced to a single number by calculating the expected value. The model can be interpreted as a probabilistic lookup table or analog model, and for applications where no analog situation was included in the training data, we set the model prediction to be a uniform - i.e. minimally informative - distribution of the output value. By design, the model cannot account for memory effects, such as those caused by water storage in the catchment.

The only way such memory effects can enter the prediction is through the model input. We therefore built several models with different sets of predictors, including those with temporal aggregations, and then selected the set of predictors that had the best predictive performance across all catchments. Notably, this is not necessarily the case for the predictor set with the largest number of variables, as over-fitting quickly occurs in such cases. We tested all possible combinations of the following options: Splitting the range of values of each variable into 2, 4, 6 or 8 bins; providing precipitation input either spatially lumped

or split into two sub-basins; providing precipitation input either as a single variable with the value of the current day, or as four variables: daily value of the current day and day -1, precipitation sum of day -2 through -6, precipitation sum of day -7 through day -30, thus providing precipitation memory; providing spatially lumped temperature as a single variable with the value of the current day, or as two variables: daily value of the current day and mean temperature of day -1 through day -30, thus providing temperature memory. Among all variants and across all catchments, the best input combination was the spatially

lumped combination of precipitation and temperature, both with memory (preceding day and preceding week), splitting the value ranges into two bins each. This model was used for all further investigations.

**Regression tree (RTREE)** Like the EDDIS model, regression trees are simple and completely agnostic to the structure of the real-world system, so their predictive power depends entirely on the information content of the chosen input data. The RTREE therefore also serves as a lower model benchmark, but it is slightly more sophisticated than EDDIS: through supervised

learning, it optimizes the partitioning of the input data to maximize the predictive power of the output. We used the "fitrtree" function in Matlab R2024a to fit the trees, testing the same input variants as for EDDIS. Interestingly, the same spatially lumped precipitation-temperature input set with memory as for EDDIS showed the best performance, and was therefore used for all further studies. Regression trees have been applied to hydrological problems e.g. by (Zhang et al., 2018; Paez-Trujilo et al., 2023).



**Artificial neural network (ANN)** The ANN consists of multiple layers of interconnected nodes or neurons, including input, hidden, and output layers. Each neuron in the hidden layer applies a weighted transformation to the input data, followed by a nonlinear activation function to capture nonlinear relationships. During training, the model adjusts its weights using backpropagation, an optimization algorithm designed to minimize the error between predicted and observed outputs. This allows the model to learn from the data and improve its predictions over time. In hydrological modelling, ANNs are used

because of their ability to capture complex, nonlinear relationships between variables (Hsu et al., 1995). However, because an ANN lacks inherent memory or recurrence, it cannot alone account for temporal dependencies in hydrological data. To account for the strong autocorrelation typically present in such data, it is necessary to shift the inputs over a time window. By applying a time window lag, the ANN can account for delayed effects, i.e., inputs from previous time steps are used to predict current conditions. In this case, the ANN is used to predict discharge based on past time series data, including variables such as

precipitation, temperature, and evapotranspiration. The input data is shifted by 7 daily time steps, generating 21 input features. The model architecture consists of 3 layers of 64 hidden units each. The first two layers use a rectified linear unit activation function. The training optimization includes a learning rate of 0.001, which decays by a factor of 0.5 after every 5 epochs. A 40 % dropout rate is applied to prevent overfitting. The model is trained for 30 epochs with a batch size of 32. To account for variability due to random weight initialization, each model is initialized and trained three times.

**Long Short Term Memory Network (LSTM)** Long Short-Term Memory networks (LSTMs) have become the benchmark model for streamflow and rainfall-runoff modelling (Kratzert et al., 2018; Acuña Espinoza et al., 2024). Unlike ANNs, which inherently cannot capture temporal dependencies, LSTMs are specifically designed to handle time-series data through their internal memory cells and gating mechanisms. In this study, three LSTM networks are built for each of the three test catchments. The model architecture consists of an LSTM layer, followed by a linear output layer, both featuring 64 hidden units. The

networks are trained with a learning rate of 0.01, and a learning rate decay factor of 0.5 is applied after every 5 epochs to optimize training. A 40 % dropout rate is used to prevent overfitting. Training is performed over 20 epochs with a sequence length of 365 days, and the forget gate bias is set to 1 to facilitate the learning of long-term dependencies. The LSTMs predict discharge using the same input features as the ANN models, including precipitation, temperature, and evapotranspiration. However, unlike ANNs, the inputs are not shifted over time. To account for variability due to random weight initialization,

each LSTM model is initialized and trained three times and we use the average of the three models in any further analysis.

### 2.3.3 Data used by models

All models are provided with the same meteorological forcing data as well as the daily discharge observations at the outlet of each catchment. Although the meteorological data was provided to each model as the same daily grid, different aggregations were applied to use the data. While the mHM model is provided with the original grids, SWAT+ uses averages of the internally

generated sub-basins and HBV and the GR4J models use sub-basins averages delineated at the gauging stations (Table 3). EDDIS and RTREE use catchment-averaged data, ANN and LSTM use sub-basin averaged data, all of them with several temporal aggregations (details are explained in the respective sections above).



To set up the process-based models, some of them make use of additional data, which allow building the specific model architecture and partly also the model parameterization. For example, SWAT+ uses soil information and land use to define soil storage and root depth (Table 3). These additional data also require different spatial discretization, e.g., for SWAT+ to the HRU and for mHM to each grid cell. For this study, these additional data are considered as part of the model structure and not as comparable input data, i.e., we treat these additional data as model-specific prior knowledge that constitutes the model architecture. EDDIS, RTREE, ANN and LSTM do not apply additional static data.

**Table 3.** Input data and temporal and spatial discretization of the data as used in the process-based and data-driven models. DEM = digital elevation model, pET potential evapotranspiration.

| Data | HBV | GR4J | mHM | SWAT | EDDIS | RTREE | ANN | LSTM |
|------|-----|------|-----|------|-------|-------|-----|------|
| DEM | elevation zone | sub-basin | grid | grid | - | - | - | - |
| Slope | - | sub-basin | grid | grid | - | - | - | - |
| Land cover | - | - | grid | HRU | - | - | - | - |
| Soil type | - | - | grid | HRU | - | - | - | - |
| Precipitation | sub-basin, daily | sub-basin, daily | grid, daily | sub-basin, daily | lumped, daily | lumped, daily | sub-basin, daily | sub-basin, daily |
| Temperature | sub-basin, daily | sub-basin, daily | grid, daily | sub-basin, daily | lumped, daily | lumped, daily | sub-basin, daily | sub-basin, daily |
| PET | sub-basin, daily | sub-basin, daily | grid, daily | sub-basin, daily | lumped, daily | lumped, daily | sub-basin, daily | sub-basin, daily |
| Discharge | daily | daily | daily | daily | daily | daily | daily | daily |

## 2.4 Distance measures and objective functions

In this section, we describe the distance measures used to address research questions Q1-Q4, specifically for data-driven catchment characterization, for model parameter estimation during model training and for model performance evaluation.

In particular, for the characterization of catchments based on available data, we needed a measure that would allow the integration of multivariate data of different dimensions on a single scale, the measurement of the total variability of catchment dynamics both with and without memory, and the direct comparison of joint unconditional variability of all variables with conditional variability of the target variable, discharge, given all other variables. All of these requirements are met by joint entropy $H_j$, an information measure, as it operates on probabilities of variable values rather than on the values themselves. A good general introduction to information theory is provided by Cover and Thomas (2006), an overview on applications in the Earth Sciences by Kumar and Gupta (2020), and a comparison to other methods of uncertainty quantification by Abhinav and Rao (2023). Recent applications of information concepts to hydrology include, among others Jiang et al. (2024a) for model training, Ehret and Dey (2023) for system classification, Moges et al. (2022) for data analysis, Azmi et al. (2021) for



model evaluation, Ruddell et al. (2019) for model diagnostics, Neuper and Ehret (2019) for hydrometeorological data-driven modelling, and Nearing et al. (2018) for process diagnostics.

Information measures exist for both continuous and discrete distributions. Computing continuous information measures typically requires fitting a continuous parametric distribution function to the data, which can be challenging, especially for high-
dimensional distributions and sparse data. Computing discrete information measures from continuous data requires binning, which inevitably leads to information loss, but is straightforward even for high-dimensional and sparse data. Since a central question of this paper is how well models learn from few data, we explain discrete information measures below and use them throughout the study.

For a multivariate set of discrete variables X1, X2,..., Xn, their overall joint variability can be measured by the entropy of
their joint distribution $H_j$ according to Eq. 1.

$$H_j(X1, X2, ..., Xn) = -\sum_{x1 \in X, x2 \in X2, ..., xn \in Xn} p(x1, x2, ..., xn) \log_2 p(x1, x2, ..., xn) \tag{1}$$

If the log of the probability $p$ is taken to base 2, $H_j$ is measured in bits and can intuitively be interpreted as the number of binary (Yes/No) questions that would need to be asked to correctly guess a particular multivariate measurement if the joint distribution were known. Entropy therefore is a measure of uncertainty expressed as number of questions. $H_j$ is non-parametric,
seamlessly expands from uni- to multivariate datasets, and is therefore well-suited for our task. Additionally, lower and upper bounds of $H_j$ exist. This allows standardization of results and thus facilitates inter-comparison between datasets of different dimensionality. The lower bound of zero is reached by a Dirac distribution, the upper bound of log2(n) is reached by a uniform distribution, where n is the number of bins.

While $H_j$ measures the unconditional overall variability of a data set, to evaluate model performance we need to measure
how uncertain we are about the value of the target variable of interest, knowing the model prediction. This is measured by conditional entropy as shown in Eq. 2, where Y is the observed target value, and X the related model prediction.

$$H_c(Y|X) = -\sum_{y \in Y, x \in X} p(y|x) \log_2 p(x) \tag{2}$$

For simplicity, Eq. 2 is shown for the case of a single target variable and a single prediction thereof, but $H_c$ like $H_j$ expands seamlessly to multivariate targets and predictions, and like $H_j$ it is bounded. The lower bound is zero, which is reached when
the model unambiguously identifies the target observation, the upper bound is the unconditional entropy of the target $H_j(Y)$, which is reached when the model has no predictive power at all.

As mentioned above, binning of continuous data inevitably loses the information about the position of each variable value within the bin, and the fewer and wider the bins, the higher the loss. On the other hand, choosing many narrow bins leads to sparsely populated and hence non-robust distributions, especially for high-dimensional data sets. Binning therefore is a two-
sided optimization problem. We solved it by choosing, for a given number of bins, their edge positions such that they i) cover the entire value range of the data and ii) minimize the sum of squared errors (SSE) between the original and the binned data,





where the latter is represented by the respective bin centre. Such an optimization is essentially a clustering problem with SSE as the measure for within-cluster distance, and we implemented it with the "clusterdata" function in Matlab R2024a. Such a binning by optimization respects both the values and the frequency of the data. Regarding the choice of the number of clusters:

For the predictors in the EDDIS model, we tested several options ranging from two to eight (see related paragraph above); for the observations and predictions of the target variable discharge then, we chose twelve as the best trade-off between resolution and bin population.

In summary, we used joint entropy and conditional entropy for data-driven catchment characterization (Q3). For performance evaluation of all models in Q1, Q2, and Q4, we used conditional entropy of observed discharge given simulated discharge. Ad-

ditionally, we provide performance results measured by the Kling-Gupta efficiency KGE (Gupta et al. (2009)) in the appendix, because it is widely used in hydrology and thus facilitates the interpretation of results for hydrologists. Several measures were used for model training. The reason for this is that the models used in this study cover a wide range from data to process based, and different training methods and appropriate performance measures are used in the respective communities. In order not to disrupt well-established method-measure interactions, we decided to keep the domain-specific measures, acknowledging

the slight inconsistency we introduced. In particular, for all process-based models KGE was used as objective function during training, for RTREE the Root Mean Square Error (RMSE) and for ANN and LSTM the Mean Squared Error (MSE) was used, EDDIS did not require any training.

In order to better emphasize how well a particular model can learn from data of a particular catchment, we introduce a standardized measure of "relative learning" as shown in Eq. 3,

$$L_{rel} = \frac{(L_m - L_{lower})}{(L_{upper} - L_{lower})} \tag{3}$$

where $L_m$ is the learning of the model defined as the difference between the conditional entropy of the model prediction when the training sample size is minimal, and the conditional entropy of the model prediction when the training sample size is maximal. $L_{lower}$ and $L_{upper}$ serve as upper and lower benchmark for standardization, inspired by the general benchmarking suggestion of Seibert (2001). $L_{lower}$ is the smallest possible value $L_m$ can take (here: zero), and $L_{upper}$ the largest (here:

the unconditional entropy of observed discharge of each catchment). $L_m$ thus takes values between minus infinity and one, where negative values indicate that the final performance is less than the lower benchmark, "zero" indicates that a model cannot learn anything from available data, and "one" indicates that a model can learn all information contained in available data, and perfectly predicts the target.

## 2.5 Experiments

For all experiments (overview: Table 4), we provide the models with ten different sample sizes from the available discharge data up to the full length of the time series. These are 2, 10, 50, 100, 250, 500, 1000, 2000, 3000, and 3654. For each sample size, the models are trained/calibrated on discharge and only the data of the specific sample size were evaluated during the training. The parameter ranges that were defined for each model can be found in the supplementary material.





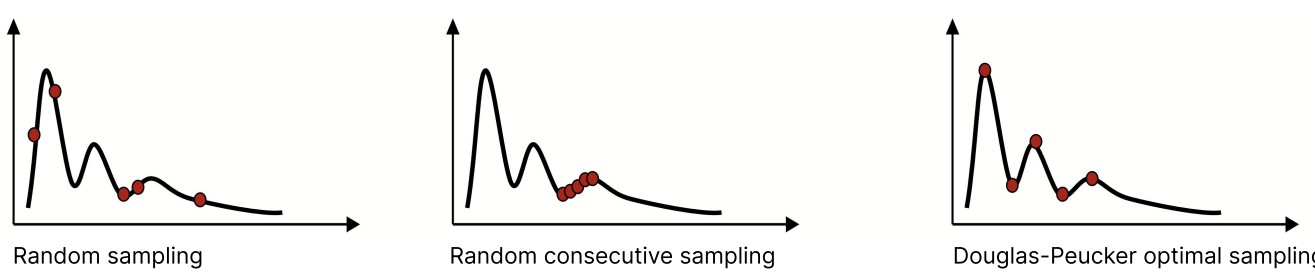

**Figure 2.** Sketch of the different sampling strategies, fully random, random consecutive, optimal sampling using the Douglas-Peucker algorithm.

For the different experiments we used various sampling schemes (Figure 2): In the **fully random sampling scheme**, we
sample $x$ random points that form the basis for calculating the model performance with $x$ being the respective sample size. For each sample size, we performed 30 repetitions, i.e., 30 random samples over the training period. For the **random consecutive sample scheme**, we randomly sampled a single point in the time series and then used all the subsequent points. If the sample size was larger than from that point to the end of the data, the points previous to the single sampling point were also used to achieve the desired sample size. For each sample size, we used 30 repetitions, i.e., we sampled a random starting point of the
continuous series 30 times. This sampling scheme resembles the case where a measurement campaign is started (randomly in time) and continues until the study or funding ends. This is probably the most common type of data set we have available for model training, and it neglects potentially interesting periods, floods, long dry spells leading to droughts.

In order to achieve **optimal sampling**, the algorithm proposed by Douglas-Peucker (Ramer, 1972; Douglas and Peucker, 1973) was selected. The algorithm searches for the most informative points, specifically targeting turning points such as flood
peaks, points before the start of the rising limb of the hydrograph and so forth. The most informative points for this algorithm are those where there are changes in the time series. This approach could be used if we did training but wanted to reduce the dimension/data size for one reason or another. An example of the points selected using the Douglas-Peucker algorithm is shown in the supplementary material for some of the used sample sizes of the Iller catchment (Figure S1).

Training is done for each sample size, each replicate, each process-based model, and each catchment using Latin Hypercube
Sampling (LHS). All models, i.e., data-driven and process-based, have a warm-up run in the period from 1 January 2000 to 31 December 2000 and all sampled training points are from the period 1 January 2001 to 31 December 2010. Model performance was validated using an independent period from 1 January 2012 to 31 December 2015, also preceded by a warm-up period from 1 January 2011 to 31 December 2011. The simulations from this validation period are the basis for all presented model performances and model learning behaviour. For details on the LHS settings, see the Supplementary Material.



**Table 4.** Overview of the model experiments, purpose, and models used. Semi-distributed means the spatial distribution that is commonly used for each model, i.e. for the HBV model divided into sub-basins, for SWAT+ in HRUs.

| Experiment | Spatial discretization | Sampling | Models |
|---|---|---|---|
| Experiment 1: How well do different models learn from limited discharge data, and more specifically is there a break point between process-based and data-driven models? | lumped, semi-distributed, distributed | random consecutive | all |
| Experiment 2: How does the strategy of selecting training data affect model performance? | semi-distributed | random consecutive, fully random, optimal (Douglas-Peucker) | HBV |
| Experiment 3: Does analyzing the information content of catchment data allow predictions about performance of different model types? | lumped | random consecutive | - |
| Experiment 4: Do spatially distributed data contain relevant, general information that goes beyond lumped data (Q4)? | semi-distributed, lumped | random consecutive | HBV |

### 2.5.1 Experiment 1: How well do different models learn from limited discharge data, and more specifically is there a break point between process-based and data-driven models?

In this main experiment, we compare the learning behaviour of four process-based and four data-driven models. We trained all models with data from the same time period using the random consecutive sample scheme, providing each model with the same increasing sample and the same thirty repetitions of each sample size to train. Although the exact same gridded data was provided for each model, different pre-processing was required for to force the models. For the semi-distributed process-based models, the data were spatially aggregated to sub-catchments (HBV model, GR4J, SWAT+), and for the simpler data-driven models (EDDIS, RTREE) they were fully lumped. By comparing the conditional entropy of each model for the independent validation period, we can assess how well the models learn relative to each other and how this changes for the different catchments for which the experiment is conducted.

### 2.5.2 Experiment 2: How does the strategy of selecting training data affect model performance?

In this experiment, we test the effect of different sampling schemes on the performance of the HBV model. To assess the influence of the training data on the actual training, we tested three different sampling schemes: fully random sampling, consecutive random sampling and optimal sampling using the Douglas-Peucker algorithm. As an example of a model commonly used in lumped or semi-distributed model setup, this experiment was performed using the HBV model.



**Table 5.** Comparison of input data characteristics for the process-based HBV, GR4J, mHM and SWAT+.

| Data source | HBV | GR4J | mHM | SWAT |
|---|---|---|---|---|
| Digital elevation model | elevation zones | sub-basin average | grid | grid |
| Slope | not included | sub-basin average | grid | HRU average |
| Land use data | not included | not included | grid map(s) for different years | one grid map, updates possible |
| Soil data | not included | not included | grid | vector |
| Precipitation data | daily, sub-basin | daily, sub-basin | daily | daily |
| Temperature data | daily, sub-basin | daily, sub-basin | daily | daily |
| Pot. evapotranspiration data | daily | daily, sub-basin | daily | daily |

### 2.5.3 Experiment 3: Is there a relationship between the information contained in the data and the shape of the learning curve for different models that allows predicting the achievable model performance?

Another interesting question that can be explored using the power of information theory, is whether we can infer the ease or difficulty of training a model from a prior analysis of the available multivariate input data (precipitation, temperature, evapotranspiration). In other words, is the resulting model performance predictable from the joint entropy of the input data? For each catchment we calculated the joint entropy of the spatially lumped variables and then the conditional entropy, given the discharge. These two types of entropy give different insights; joint entropy about the overall information content of the data and conditional entropy about the information content that is relevant to discharge conditions.

We computed the unconditional entropy of the data sets with four variables (P, T, PET and Q) with and without memory, i.e. with and without any temporal dependence of the data: 1) only the value at time step $t$ is used, resulting in four variables (P, T, PET, Q); 2) in addition to the four variables at time step $t$, the variables at time step $t-1$ are also used, resulting in eight variables that are each binned; 3) in addition to the four variables at time step $t$ also the variables averaged over the preceding week $t-1$ to $t-6$ is used, resulting in eight variables that are each binned. The joint entropy values between cases 2) and 3) can be directly compared, since the base are both times eight variables, each in eight bins. Instead, the values of 1), which does not take into account any memory, cannot be directly compared to 2) and 3), since the base is only four variables, each in eight bins (Table 6).

We computed the conditional entropy of the data with respect to the target variable discharge Q at $t0$ based on the three predictor variables P, T and PET without memory, i.e. also at $t0$. Again, we looked at the three cases 1) three predictors and no time aspect is considered at all, 2) three predictors and for each variable also the value of the preceding day, 3) three predictors and for each variable also the average value of the preceding week (Table 7). These conditional entropy values can be compared with each other, as here the discharge was binned into eight bins for each analysis. The maximum entropy value for these cases is 3 (= log2(8)).





### 2.5.4 Experiment 4: Do spatially distributed data contain relevant, general information that goes beyond lumped data (Q4)?

To test the benefit of more spatial distribution in the input data for model training, we used the HBV model and set it up for
each catchment in both a lumped and in a semi-distributed manner by using sub-catchments. Both model versions were trained
using the same time periods and the consecutive random sampling scheme, but the lumped model received catchment areal
averages of precipitation, temperature and evapotranspiration, while the semi-distributed models received these meteorological
inputs averaged to the sub-catchments.

## 3 Results and Discussion

### 3.1 Experiment 1: How well do different models learn from limited discharge data, and more specifically is there a break point between process-based and data-driven models?

From this experiment, we were able to generate learning curves for each model and each catchment. The learning curves show
how much a model has learned with increasing sample size during training. This means that if the conditional entropy, Hc,
decreases with increasing sample size, the models are learning from more discharge data.

For all catchments we found a grouping of the process-based models and the data-driven models, where the data-driven
models learn longer than the process-based models, but the process-based models start with lower conditional entropy values
than the data-driven models. However, this is expressed to different degrees for each of the three catchments.

The learning curve for the Iller catchment (Figure 3, left panel) shows that all models decrease the conditional entropy, Hc,
i.e. they learn with increasing training sample size. This is true for all models, both data-driven and process-based. The process-
based models start with lower conditional entropy values (between 2.3 and 1.8) when provided with very small sample sizes
than the data-driven models (Hc 2.1 to 2.6). The process-based models learn with increasing sample size and reach a learning
plateau at around 500 samples. The learning curve for the mHM model is very flat, showing almost no improvement with
increased sample size, but is also surrounded by a very wide band from the repetitions around the median learning curve. The
data-driven models continue to learn with increasing sample size, and the LSTM in particular outperforms all the process-based
models. The ANN also continues to learn and reaches performances that are comparable to some of the process-based models.
The simple data-driven models (EDDIS and RTREE) have a steep learning curve at the beginning and almost the conditional
entropy value reach the performance of the process-based models SWAT+ and mHM. However, compared to the LSTM, their
learning is much slower after a sample size of 500.

For the Saale catchment (Figure 3, middle panel) there is a similar grouping of process-based versus data-driven models.
Again, the process-based models start their learning curve at lower conditional entropy values than the data-driven models
and soon reach a plateau with a relatively small sample size. The data-driven models have a very steep learning curve at the
beginning and continue to learn slowly, but they do not reach as low conditional entropy values as seen in the Iller catchment,
nor do they reach the performance of the process-based models. Three models of the model groups stand out. Among the





process-based models, the SWAT+ model has a lower performance at the beginning of the learning curve, i.e. when it is provided with a small sample for training, and it remains in the performance below the other process-based models until the end of the learning curve, i.e. when it is provided with all the data available for training. The ANN model starts with a rather high performance compared to the other data based models and stays in a similar arrangement to the learning curve of the process-based models. The LSTM, as already seen for the Iller catchment, has the steepest learning curve and continues to learn after all other models have finished. For the Saale catchment the learning curves were quite different from those of the Iller catchment, and all models, process-based and data-driven, except for the LSTM model plateau after a sample size of 500 to 1,000. This catchment showed an intermediate data variability and an intermediate learnability from the ranking of the joint entropy of the input data and the ranking of the conditional entropy.

For the Selke catchment (Figure 3, right panel), there is a relatively wide spread between the learning curves of the process-based models with the simplest models GR4J and HBV starting and reaching the highest performance, although the overall learning curve is rather flat. The SWAT+ and mHM models both start with a similar performance, but essentially show no learning with increasing sample sizes larger than 500. The performance of the simple data-driven models (EDDIS and RTREE) and ANN is very similar to the SWAT+ model. The simple data-driven models start their learning curve with very low performance, learn quickly with increasing sample size and stop learning at a sample size of 1,000. The ANN model continues to learn, but with a very flat learning curve. The LSTM, like for the other two catchments, knows very little at the beginning, then has a steep learning curve and continues to learn. Unlike for the Iller and Saale catchments, the LSTM does not outperform the process-based models HBV and GR4J.

If we compare only the relative learning (Figure 4), i.e. how much a model learns without considering the initial performance, but only comparing the performance at the beginning and at the end of the learning curve, then we see that the data-driven models learn the most, with the LSTM model clearly outperforming all other models. For this consecutive sampling scheme, the HBV model also shows a learning capability comparable to the ANN and RTREE. However, it should be noted that it does so mainly in the first sample size increase and then plateaus (Figure 3). For our comparison the SWAT+ model has a lower learning capability, and for the GR4J model, the learning capability varies with the catchment in which it has to learn. The mHM model shows the least learning with almost no response to sample size irrespective of the initial performance. Additionally, there is a large band around the median learning curve, indicating large parameter uncertainty at every sample size.

It is well known that discharge is much easier to model in some catchments than in others, either because of a pronounced seasonality in discharge expressed as spring flood, summer low flow, or the like, or because the catchment response is very similar to the precipitation that falls on the area or also because there may be errors both in the forcing meteorological data and in the data used for evaluation, in our case discharge. These errors can be so large that, as a consequence, the preservation of mass (inscribed in process-based models but not for data-driven models) is violated and the water balance is not closed. What is interesting, however, is how differently the process-based models we used in our study were able to simulate and learn the discharge with the same input data. For example, the Iller catchment, which had a high variability but also a high learnability from the data alone, has a high spread of the learning curves of the individual models, indicating that the model





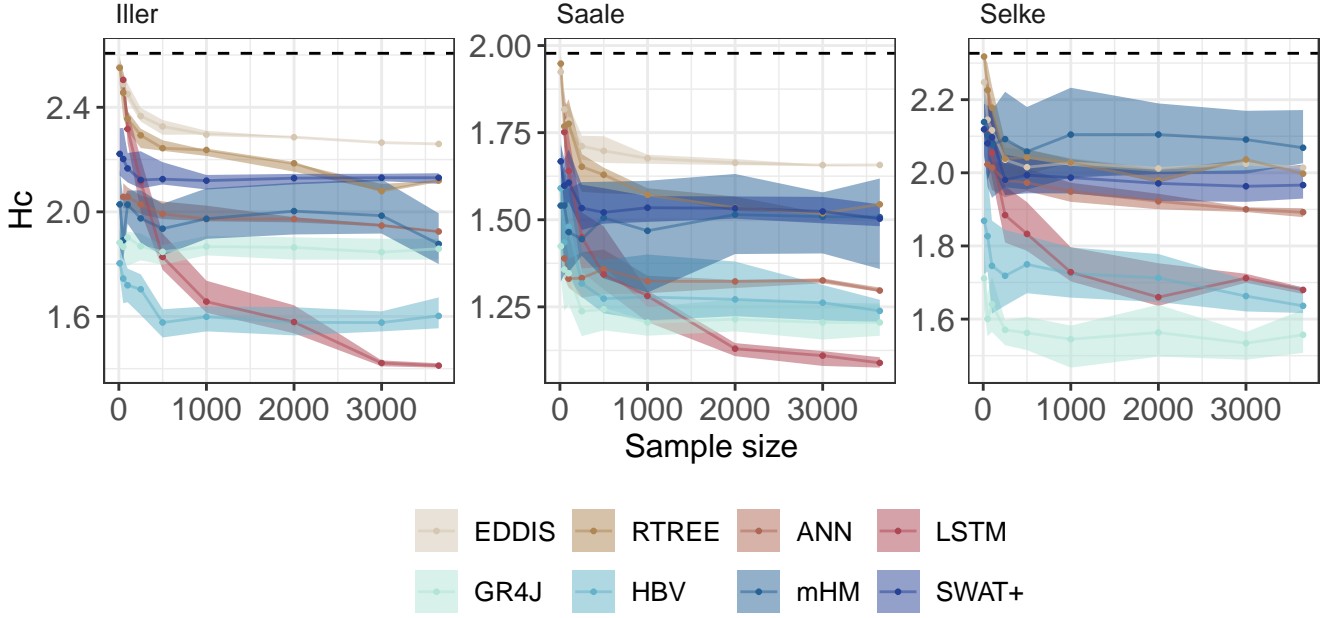

**Figure 3.** Learning curve using the continuous random sampling strategy for the different models and catchments, conditional entropy, Hc. The lower the values of Hc, the more the model could learn from the data (i.e., the better discharge simulations are). The band of the learning curves are the 25th and 75th percentile of the ensemble of 30 repetitions, the line is the median. The dashed line shows the maximum possible entropy. Note that for visibility reasons we applied a different y-axis scaling for each catchment.

architecture of some of the process-based models is somehow more advantageous than the model architecture of others for the
training of this catchment. The Iller catchment could be modelled well and learned well (up to the learning plateau) by the rather simple process-based models GR4J, HBV, but with generally worse model performance for the spatially higher resolved SWAT+ model. The mHM model started and ended with about the same value of conditional entropy as processing along the learning curve, i.e. with increasing sample size. As the model is widely used and calibrated (Zink et al., 2016; Shrestha et al., 2024), this suggests that the learning curves obtained could be influenced by the LHS setting, which may be too small for this
model.

The strongest learning performance was observed for all data-driven models, with the LSTM showing the steepest learning curve and ultimately achieving the highest predictive accuracy once a sample size of 2,000 was reached. Unlike simpler data-driven models (e.g., (EDDIS and RTREE)), both the ANN and LSTM models used semi-distributed input data. It is likely that the LSTM benefited from this input by discovering complex relationships that simpler models could not. While the
discretization of the input data does not explain why the LSTM continues to learn when the ANN stops, this behaviour can be attributed to the model architecture itself. The LSTM, similar to a classical hydrological model, essentially operates as a state space model. We refer to this inherent architectural advantage as an inductive bias: unlike standard ANNs, which lack



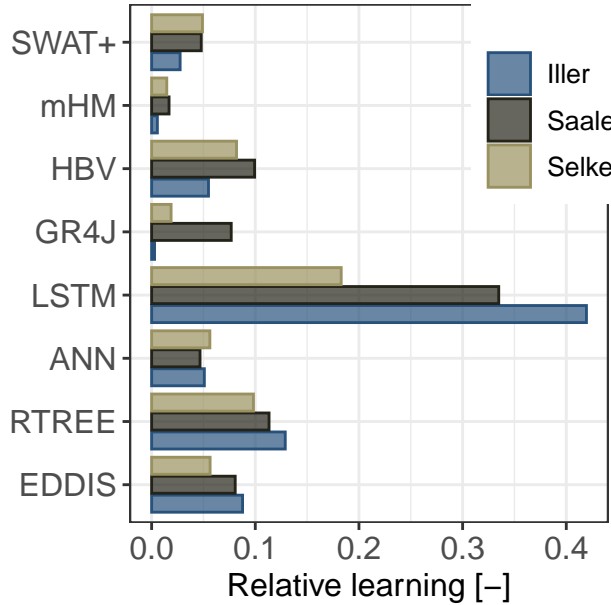

**Figure 4.** Relative learning of the different models using the continuous random sampling scheme. Relative learning is defined as the difference between the beginning and the end of the learning curve (Eq. 3).

memory cells and intrinsic recurrence, the design of the LSTM (De la Fuente et al., 2024) allows it to continuously integrate new information over time, enabling persistent learning and improved performance.

## 3.2 Experiment 2: How does the choice of training data, i.e. the information content in a given data set, affect training for a specific problem?

The different sampling strategies have only been investigated for the HBV model. Here, the learning curves for the Iller (Figure 5) show how different sampling compositions affect learning. We have a band around the consecutive random and the fully random sampling because we did 30 repetitions of the random sampling, but one line for the learning curve of the Douglas-Peucker sampling because this algorithm gave us only the most interesting points, presumably the optimal sampling. Especially, the learning curves of the random and consecutive random sampling strategies look very similar. For these two, the HBV model learns more for the Iller and Saale catchments than for the Selke catchment: When expressing learning as the difference in conditional entropy between the smallest and the largest sample divided by the conditional entropy of the smallest sample, then learning reduced conditional entropy by 10.3 percent for the Iller, 16.6 percent for the Saale, and only 9.1 for the Selke (values are averages of the random and consecutive random approach). The Douglas-Peucker learning curve appears to be much jumpier than the learning curves of the full random and the consecutive random learning curves. It should be noted that the learning curves for both random sampling schemes show the median of 30 replicates and the 25th and 75th percentiles,





smoothing out the behaviour of the individual lines used to calculate these statistics. Each of these individual repetition lines could (and some do) have jumpy behaviour similar to the Douglas-Peucker curve.

The learning curve using the Douglas-Peucker sampling starts with the highest conditional entropy for the Iller catchment, but the lowest conditional entropy for the Saale and Selke catchments compared to the other two sampling strategies. The largest samples show exactly the same entropy value in the learning curves, starting for the Iller catchment with a sample size of 1000, for the Saale with a sample size of already 500 and for the Selke with a sample size of 2000. These same entropy values are derived from the selection of exactly the same model for these sample sizes, i.e. there is no further learning with the

additional points that increase the sample size.

When using the consecutive random sampling scheme the HBV model learns approximately up to a sample size of 1000 and then reaches the plateau of the learning curve. The fully random sampling scheme gave the best performance, i.e. the lowest conditional entropy, compared to the others at the beginning of the learning curve and also plateaus around a sample size of 500. The learning curves for Iller and Saale look very similar at the end for all sampling schemes and there is no significant

difference visible in model performance in terms of conditional entropy. For the Selke, the conditional entropy of the Douglas-Peucker sampling is higher than for the two random sampling strategies, which can be explained by the additional points in the larger samples provided for training. These additional points come from turning points before the rising limb and contain more low flow values. Optimization focusing both on low and high flow (first selected by the algorithm) then attempts to optimize for both flow aspects and the overall model performance is reduced. Using a different training setup that increases the sample

size of the LHS, or using a different training algorithm such as the dynamically dimensioned search (Tolson et al., 2009), could help avoid this worsening in the learning curve (Figure S1 supplementary material). Using the KGE instead of the conditional entropy for the learning curve does not show such a decrease to lower performance in the Selke catchment and shows smoother learning (Figure S1, Supplementary material). Since we included only the HBV model in this side experiment to test different sampling schemes, we cannot make a general statement. However, with relatively small sample sizes between 500 and 1000

days the model has already learned to its maximum. Similar ranges were also found in (Brath et al., 2004; Melsen et al., 2014; Sun et al., 2017). For the catchments we used in this study, all of which are in a humid climate, it does not seem to matter whether the sample is made up of a continuous time series of discharge or a random sample over many years, suggesting that these sample sizes sufficiently represent the natural variability in flows and hydrologically active periods.

With the resulting learning curves when using random consecutive sampling, we can infer how long such a measurement

campaign should last in a given catchment to capture enough information for the model to learn. This is different for each catchment but somewhere between 500 and 2000 points, and comparing the variability inherent in the data that we could quantify using the joint entropy, the conditional entropy of the discharge using these variables as predictors and the learning curves themselves can provide useful guidance.

The other two sampling schemes we tested are fully random and optimal using the Douglas-Peucker algorithm. These

sampling strategies are more commonly used in practice during event-based sampling campaigns, or when there are large data sets and only a representative or essential information is sampled to reduce computational efforts for model training. We found that fully random sampling slightly outperformed Douglas-Peucker sampling, despite the idea that this algorithm would





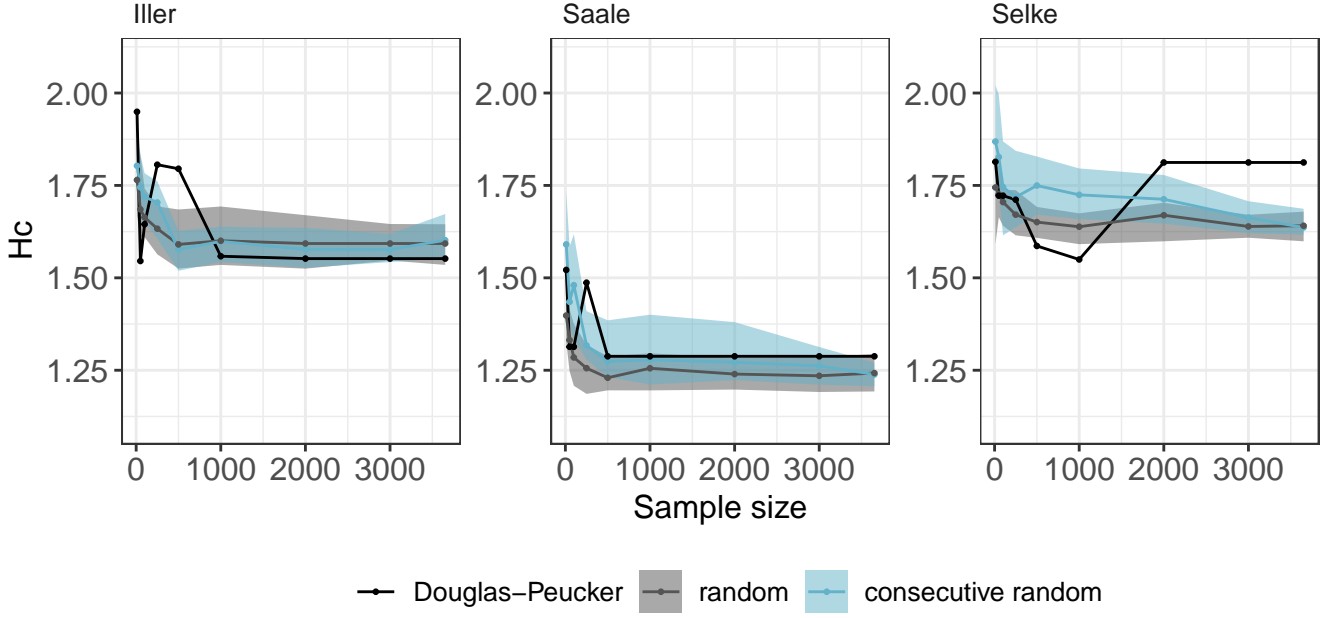

**Figure 5.** Learning curve using the different sampling schemes for the HBV model, Hc, conditional entropy. The lower the values of Hc, the more the model could learn from the data (i.e., the better discharge simulations are). The band of the learning curves are the 25th and 75th percentile of the ensemble of 30 repetitions, the line is the median.

provide the model with optimal sampling points for training. This was particularly true for the Selke catchment, where the learning, expressed as a reduction in conditional entropy, was reversed because additional points in the sample realigned the

model focus away from mainly floods to also low flows, and the parameter search ended in exactly the same model for all samples larger than a catchment-specific sample size. A major advantage of random sampling over optimal sampling is the ability to repeat the sampling, which ultimately provides learning curves from statistics that can be used as a guide, rather than overthinking why one increase in sample size did not produce the expected learning while the next did.

### 3.3    Experiment 3: Is there a relationship between the information contained in the data and the shape of the learning
550        curve for different models that allows predicting the achievable model performance?

In this experiment we focused on the information content in the data using information theory and simply taking the data of each time step to predict the discharge at that time step, but also including memory in the form of input variables aggregated over time.

    If no memory is included in the predictors but only precipitation, P, temperature, T, potential evapotranspiration, pET at time
step $t0$, then the ranking is exactly the same as we found for the joint entropy of the meteorological variables and discharge



(Table 6): the highest conditional entropy for the Iller catchment (1.74), the lowest for the Selke catchment (1.39), suggesting that there is the highest variability in the Iller, less in the Saale and lowest in the Selke catchment.

However, if we add the preceding day as information, then the entropy for all catchments decreases, indicating learning, and the ranking of the catchments changes: now, the Iller catchment has the lowest entropy (0.81), the Selke catchment has the highest entropy (0.94) and the conditional entropy for the Saale catchment is in between the other two (0.92). Adding the information of one week before $t0$ instead of one day again decreases the conditional entropy values for all the catchments and also in this case the Iller catchment has the lowest entropy, Selke the highest and Saale in between. If we look at the learnability then we see that the entropy reduction is greatest for the Iller catchment, even though it has the highest joint entropy values. The lowest reduction is found for the Selke catchment, indicating that there may not be so much gain in including a rather short memory temporal aspect to model discharge for the Selke as is found for the Iller and the Saale catchments.

The Selke catchment appears to be not very learnable despite the rather low data variability. It may be that the Selke catchment could be more learnable if the processes relevant to the catchment response were covered with adequate data. But it appears that the meteorological data as well as the time dependencies are not sufficient for any of the tested models. It would be interesting for the Selke catchment to test both with a longer-term memory and with different input data and constraints on different variables rather than just runoff, which could be useful in describing these processes (Wagner et al., 2025).

The joint entropy of the input data for each catchment can be used to describe the variability that needs to be captured by the models, but the link from the joint entropy to the learning of the data-driven and process-based models could not be made directly. Instead, the results indicated that the catchment with the highest joint entropy was in fact the one that had the best learnability, with the models learning the most by increasing the sample size. On the contrary, the catchment with the lowest joint entropy also showed the worst learnability. As we could already see with the conditional entropy of the input data to the runoff, an advantage to learning was the ability to intelligently incorporate memory. This is not surprising, as there have been many studies showing that data-driven machine learning approaches had a hard time simulating the runoff adequately until they included some kind of memory that would handle the temporal dependencies, the catchment antecedent conditions, and thereby increase the model performance tremendously (Kratzert et al., 2018; Shen, 2018; Fan et al., 2020). Nevertheless, from these conditional entropy values, we can see that by including more memory to condition Q we open up new ways of learning from the data for all catchments. Thus, if we use a model that can intelligently take into account the information that is inherent in the time component, we expect better learnability despite a potentially very large entropy in the data.

**Table 6.** Joint entropy of the data considering memory and not considering memory.

| Variables | Iller | Saale | Selke |
|---|---|---|---|
| P, T, PET, Q | 8.41 | 7.90 | 7.32 |
| P, T, PET, Q, $t-1$ | 11.18 | 10.79 | 10.21 |
| P, T, PET, Q, $t-1$ - $t-6$ | 11.78 | 11.56 | 11.20 |



**Table 7.** Conditional entropy of the input data regarding discharge. With and without the consideration of memory.

| Predictor variables | Iller | Saale | Selke |
|---|---|---|---|
| P, T, PET | 1.74 | 1.59 | 1.39 |
| P, T, PET, $t-1$ | 0.81 | 0.92 | 0.94 |
| P, T, PET, $t-1$ - $t-6$ | 0.55 | 0.63 | 0.69 |

## 3.4 Experiment 4: Do spatially distributed input data carry relevant information and thus enhance learning without compromising the generality of what has been learned?

The results of our experiment of changing the spatial discretization of the model input data for the HBV model (Figure 6) showed that for the Iller and the Selke catchments the model performance is generally, i.e. for all training sample sizes, much better when the model input is semi-distributed rather than lumped. For the Saale catchment there is also a slight performance improvement when using the semi-distributed model input, but the benefit is more pronounced for smaller training sample sizes and disappears for the larger sample sizes.

For the Saale catchment, there was essentially no improvement when using the semi-distributed input compared to the lumped input. Here, the sub-basins are more similar in terms of catchment characteristics such as topography and geology but also in terms of precipitation inputs to each other than in the Iller catchment. Therefore, the use of the lumped catchment average does not imply a great loss of information regarding the input data P, T and PET.

For the Iller catchment there was an offset in the performances of the HBV model, when provided with both semi-distributed
and lumped input data. This offset can be explained by the different sub-basins of the Iller, which cover different elevation zones. This implies that also the variability of precipitation, which is related to the altitude, is significantly different from the lumped input. The same is true for temperature, which in the semi-distributed input is more likely to simulate more realistic snow accumulation and melt and seems to have a positive effect on the model performance. For the Selke catchment, there is one sub-basin that is higher than the rest of the catchment and receives more precipitation than the other two sub-basins,
and resolving this more closely in the semi-distributed input may explain the gain in model performance when using the semi-distributed input rather than a lumped catchment average.

While the shape of the learning curves for the Iller catchment is similar, the shapes of the semi-distributed versus the lumped HBV model inputs for the Selke catchment are not. Here it appears that not only the performance improves, but that there is also an improved learning both for the small sample sizes and still for the larger sample sizes. The learning curves of the HBV
model in the Selke catchment are again with better performance throughout and show this slight learning advantage when using the semi-distributed input. There is a sub-basin that is very different from the rest of the lowland in terms of elevation, and accounting for this in the input might help the model to better capture the dynamics. The better learning compared to the lumped input suggests that specifically accounting for this sub-basin and its variability provides useful information in the additional data with increasing sample size that would be smoothed out for the lumped input.




The benefit of a more spatially explicit input to the HBV model has been previously investigated for other regions. Lopez and Seibert (2016) found improved model performance (Nash-Sutcliffe efficiency), but as we also found, the improvement was site-specific and very variable for a pre-Alpine region with a strong climatic gradient in Switzerland. Huang et al. (2019) looked at four catchments in Baden-Württemberg, Germany, and found only marginal model improvement with higher spatial discretization of the input data, but in their study the higher spatial discretization came from resolving elevation zones rather
than sub-catchments.

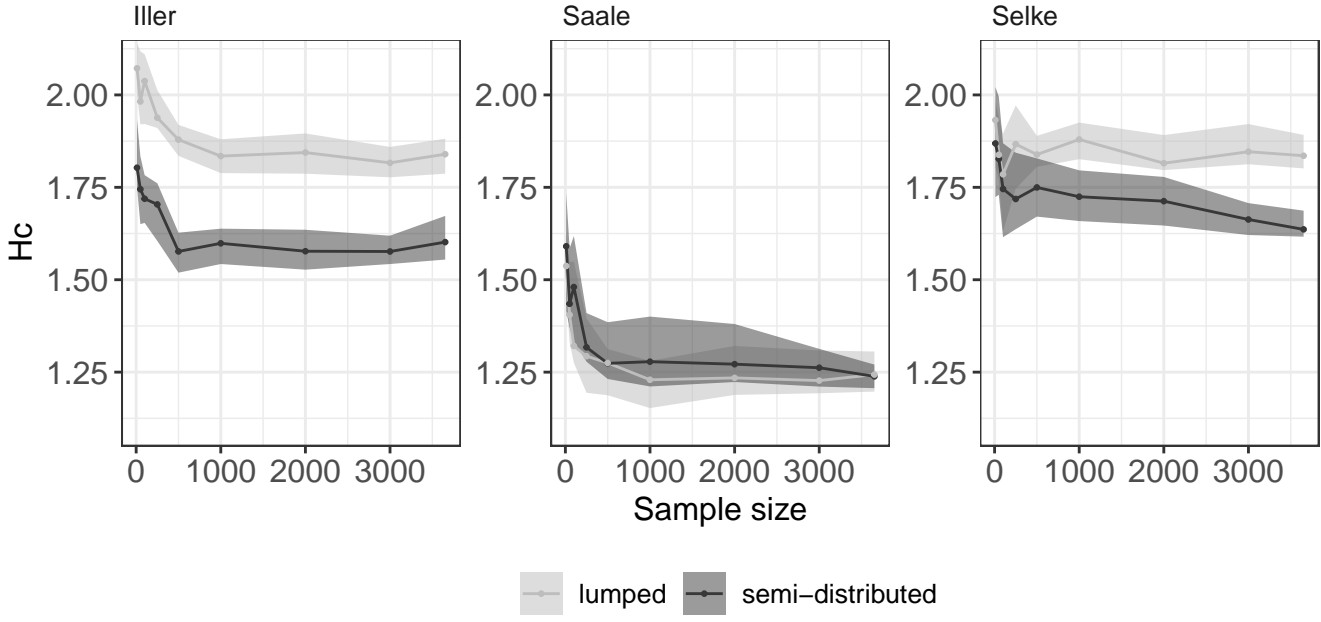

**Figure 6.** Learning curve using different spatial discretizations of the forcing data for the HBV model, Hc, conditional entropy. The lower the values of Hc, the more the model could learn from the data (i.e., the better discharge simulations are). The band of the learning curves are the 25th and 75th percentile of the ensemble of 30 repetitions, the line is the median.

## 3.5    Limitations of the study design

There are several limitations in the study design, mainly due to choices made explicitly for the experiment, but also due to model-specific constraints and feasibility.

We chose a consistent LHS for all models (see details in Supplementary Material) regardless of the different number of
parameters. However, if the Latin Hypercube sampling is too small for the parameter space spanned by the parameters of a (hydrological) model, this can lead to significant limitations. A small sample size may not adequately capture the variability and complexity of the input parameters, resulting in biased or incomplete representations of the model's behaviour. This can generally lead to underestimation of uncertainties and also impact the interpretation of the learning curves. For example,





for the mHM model, parameter sets that can be directly applied in a default setting have higher model efficiencies for our catchments than those found from the LHS sampling over the given model ranges (Supplementary Material). Therefore, the interpretation of the model learning, especially for models with a larger number of parameters, needs to be done carefully. Another approach would be to derive learning curves from the parameter samples drawn by a gradient-based optimization algorithm (e.g., Shuffled-Complex Evolution). This approach would have the challenge that it needs to be consistently applied to all investigated methods.

The data are not exactly the same for all models, although we have tried to make them as similar as possible in this framework. Part of the difference in the learning abilities of the models could be explained by the different discretization used to form the actual model forcing from the exact same gridded meteorological data provided as input for all models. However, the question of which model was the best learner - apart from LSTM, which clearly stood out - could not be answered directly from the discretization used for each model. Instead, we found that this was different for the three study catchments.

We argued that the additional data used by some models, such as global soil maps for the mHM, or the soil types and land use types used in the mHM and SWAT+ but not in the simpler process-based models and not in the data-driven models, are part of the model itself and could therefore be considered as the model architecture itself. It may be that these additional data are beneficial to model performance, but this was not evident from our results. The soil information for mHM may even have caused the model to show the small affinity to learn compared to all other models. The information inherent in the model may have prevented the model from making use of the additional discharge data provided and thus hindered the learning process.

It should also be noted that there is uncertainty in the data and that measurement or interpolation errors have not been explicitly investigated in this study. Certain types of models can deal with this in the sense that they would adapt to the data provided to them. For example, data-driven models would still find a statistical relationship between meteorological input and discharge, even though parts of the data may contain substantial errors. For the process-based models, the model structure does not allow such a high degree of flexibility and this may be reflected in poor model performance. Within the family of process-based models, the less complex models that were designed to focus on runoff prediction such as HBV or GR4J, can still provide a rather flexible way of attempting to model the input-response relationship through the choice of model parameters, whereas the more complex process-based models with higher spatial resolution, focusing on different hydrological processes within the catchment and using runoff as a means of evaluating the model, have much less flexibility. This means that they will not perform well if the data used to force and evaluate the model is error-prone.

We have studied only three catchments in Germany. These catchments are different from each other, but more in terms of topography and local climatology than in terms of different climate zones. We chose these example catchments in order to be able to find some explanations for the different learning of the models, the different data variability and the learnability. The influence of the elevation included in the model and also the processes that most influence the catchment response are probably represented using these three catchments, at least for the catchments in Central Europe, i.e. high elevation catchments with snow influence, hilly mid-mountain catchments and catchments with a high lowland coverage. Some of the results are probably site specific and not transferable to other catchments. For example, the Selke catchment has a very distinct sub-basin with a steep topography, which is not present in all lowland catchments, and the semi-distributed data may not help learning



compared to a lumped data input to the HBV model. The methodology could be applied to a larger set of models, focusing
from a large sample point of view on how much variability correlates with learnability. However, using the three catchments
allowed a more detailed look at each of them.

Learning in our study setup is limited to constraining and evaluating to discharge and no other variables such as evapotranspiration or groundwater table. The results presented may change using other and additional variables to evaluate the learning.
We would not expect a huge change in the general learning behaviour when comparing data-driven and process-based mod-
els, but a change in the shape of the learning curve with a general slowing down of the learning rate. How the ranking of the
different process-based models would be affected cannot be answered here, but would be interesting to investigate in the future.

### 3.6 Information theoretical measures in hydrological studies

Information theory can be a powerful tool to address hydrological problems. One advantage is the dimensionless evaluation
of probabilities connected to data rather than the evaluation of their original values. This allows the variability of all data
used for modelling to be estimated in the single metric joint entropy. The study catchments and their data could be compared
with the joint entropy, showing that in our case the highest variability was found in the Iller catchment and the lowest in the
Selke catchment. In order to investigate the relation of the variability of the data and learnability of a predictive model thereof,
we compared the unconditional entropy of the data set with the conditional entropy of discharge given all other variables.
Here, the order of the catchments was reversed: The Iller catchment, despite the highest data variability, also showed the
smallest conditional entropy of discharge, i.e. had the highest learnability. The Selke catchment, on the contrary, had the lowest
unconditional joint entropy of the catchment variables, but also the highest conditional entropy, i.e. lowest learnability of the
data.

Comparing the ranking of the different models when evaluating the performance using the information theory metric Conditional Entropy and the more commonly used hydrological metric KGE, only small differences were found. However, using
the information theory metric allowed a more direct comparison with the conditional entropy we calculated to express the
learnability of the catchments.

Comparing the learning curves of different models is also very useful in terms of how much of a learner the model itself is,
despite the model's starting or ending performance and ranking in performance. What is more interesting is the learning from
start to finish and when the models stop learning.

There are issues when comparing catchments with classic hydrological performance metrics, because even though standardized or normalized in their scale, these metrics do not provide an indication of how the model performs in absolute terms
(Schaefli and Gupta, 2007). Some authors hence strongly advocate for benchmarks in hydrology that consider the catchment's
complexity and the difficulty to simulate hydrological processes in a region.(Seibert, 2001; Schaefli and Gupta, 2007; Pappenberger et al., 2015; Seibert et al., 2018; Knoben, 2024). Using information measures throughout the entire workflow of data
analysis, model training and model evaluation in hydrology could help mitigate this issue, but is currently rarely done: While in
very few studies (Jiang et al., 2022, 2024b) information measures are used for specific parts, for final model evaluation usually
well-known metrics such as KGE and NSE are used for easier interpretation by the readers.





## 4 Conclusions

In this study, we investigated how different models can make use of the information in discharge data, and what kind of data
is most useful for models. To do this, we carried out four experiments: The main experiment, experiment 1, was designed to
assess the differences in the learning capabilities of different models, four of them data-driven and four process-based, with
varying degrees of complexity. Experiments 2 and 4 were designed to answer related questions about learning with different
sampling schemes and spatial discretization of the input data. We also investigated how much this varies for different catchment
types in a humid climate, including the central German lowlands, the mid-range mountains, and the Alpine region of Germany.
In experiment 3, we investigated whether it is possible to predict how well models can deal with a given data set. For this
experiment, we used information theory to describe both the variability in all the data used by the model and the learnability,
in the sense of how much information in the data could actually be useful for a model predicting discharge, using joint and
conditional entropy.

There is a difference between how variable the data set of a particular catchment is and how easy it is to learn from it. The
perhaps intuitive notion that the more variable the data for a given catchment, the more difficult it is to learn from it, does not
hold. We also found that different models are different learners and this varies also for the catchment for which they are set
up. That means the different learners are not performing equally well for all catchments, and the ranking of which was the best
learner is changing.

In general, however, the process-based models used in our study initially know more than the data-driven models due to
their model architecture, which includes some memory capabilities and thus the ability to account for memory. While this fixed
model architecture appears to be advantageous at the beginning of the learning, i.e when only few data points are provided
for training, process-based models stop learning relatively soon and plateau at a certain model performance, i.e. after a certain
amount of data has been included. On the contrary, the data-driven LSTM model had very poor performance at the beginning
of the learning curve to then learn quickly and steadily with more data provided for training. The LSTM continued to learn
after all the process-based models stopped learning and is very useful as a benchmark learner.

Applying three different sampling schemes to provide the same sample size for training showed that a fully random sampling
provides the best basis for learning, consecutive random sampling - as it would be realistic from different sampling campaigns
over a period - reached a similar performance for a large sample size, and surprisingly the optimal sampling using the Douglas-
Peucker algorithm did not outperform the two random sampling schemes in the tested catchments.

Regarding the spatial discretization of the input data from sub-catchment to lumped, we found that reducing the spatial
discretization of the meteorological input to the model resulted in an overall decrease in performance, the extent of which,
not surprisingly, depends on the homogeneity of the catchment and, in our cases, to a large extent on the forcing data in the
different sub-catchments. We have only considered the effect of meteorological forcing at different resolutions, but other data
may be relevant for the lowland catchment, which showed the least improvement.

Joint entropy is a simple yet powerful way of estimating the variability of the data associated with a catchment, as it can
handle data that comes with different dimensions. Conditional entropy tells us how this data can be used to predict discharge.



When no memory is taken into account, the conditional entropy is large, but as soon as some memory is introduced in the form of aggregations of variables over the current and past day or past week, the conditional entropy becomes smaller, indicating that memory is a very important component of the data and that capturing it improves the model performance. This was particularly

evident in the catchment from the low mountain ranges and the Alpine region.

*Code availability.* The code to calculate the conditional entropy from the model simulations, input data and discharge data is provided through a GitHub repository https://github.com/MariStau/IMPRO_infotheory_Data_Code and via Zenodo at https://doi.org/10.5281/zenodo. 14938050

*Data availability.* This publication has been prepared using European Union's Copernicus Land Monitoring Service information; https: //doi.org/10.2909/960998c1-1870-4e82-8051-6485205ebbac. The digital elevation models of the catchments were retrieved from http:// hydro.iis.u-tokyo.ac.jp/~yamadai/MERIT_Hydro/ (Yamazaki et al., 2019). The model input data and results is provided through a GitHub repository https://github.com/MariStau/IMPRO_infotheory_Data_Code and via Zenodo https://doi.org/10.5281/zenodo.14938050

*Author contributions.* Conceptualization and methodology: UE and MS. Data curation: AH and ST. Investigation and formal analysis: MS, UE, AH, TH, RL, JM, DS. Funding acquisition: BG. Visualization: MS and AH. Writing - initial draft: MS, UE and SP. Writing – review and editing: all authors. All authors have read and agreed to the published version of the manuscript.

*Competing interests.* Some authors are members of the editorial board of HESS.

*Acknowledgements.* We acknowledge funding by the German Research Foundation for the scientific network on Identification and analysis of process limitations in hydrological model structures (IMPRO, Project number 471280762). For providing the discharge data, we thank these three local authorities in Germany: LfU Bavaria, TLUBN Thuringia, LHW Saxonia-Anhalt. Simulations were performed with computing resources provided by ZIM, University of Potsdam.



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
