# Peer review of "How well do hydrological models learn from limited discharge data? A comparison of process- and data-driven models"

_EGUsphere, 2025_

## Author Response (AR1)

*Dear Nunzio Romano,*

*Thank you for your efforts with our manuscript. Please see the point-by-point response to the two reviewers' comments below.*

*As mentioned in our replies to the reviewers, please note that, in addition to the changes requested by the reviewers, we are planning to remove the mHM model from the analysis in the revised manuscript. We had already outlined the limitations of our experimental settings for models with many parameters, particularly for the mHM model (see limitations L619-629 in the original manuscript). Following further consideration, the mHM model contributors concluded that the results obtained under these settings could not meaningfully contribute to the study. Removing the mHM model does not affect the main conclusions. In the revised manuscript, we have ensured that this change is implemented consistently throughout.*

*Sincerely, also on behalf of all co-authors,*

*Maria Staudinger*

**Reviewer 1 (Salvatore Manfreda)**

**The abstract clearly outlines the study's objectives and key findings. However, the novelty of systematically comparing PB and DD models under limited data scenarios deserves more emphasis upfront. The finding that LSTM outperforms PB models after just 2–5 years of data is particularly impactful and should be highlighted earlier.**

*We rewrote to "To investigate this, we calibrated several process-based and data-driven hydrological models using training datasets of observed discharge that differed in terms of both the number of data points and the type of data selection, allowing us to make a systematic comparison of the learning behaviour of the different model types to emphasize the novelty of our approach in the abstract ."(L2-6 in the revised version of the manuscript). We also changed the description of the results here to better highlight the performance of the LSTMs already in the abstract:*
*OLD: "However, with increasing amounts of training data, the learning curve of process-based models quickly saturates, and using about 2 to 5 years of training data, the data-driven LSTM consistently outperforms all process-based models."*
*NEW: "However, as the amount of training data increases, the learning curve of process-based models quickly saturates and data-driven models become more effective. In particular, the LSTM outperforms all process-based models when trained with more than 2-5 years of data and continues to learn from additional training data without approaching saturation." (L19-21 in the revised version)*

**The mention of "conditional entropy" as a key metric is intriguing, but its relevance is not explained. A brief statement on why this information-theoretic approach was chosen over conventional performance metrics (e.g., NSE, KGE) would improve accessibility for readers unfamiliar with this concept.**

*We assume that this comment relates to the abstract. We will better justified the use of information measures by rewriting the abstract to: "We used information measures (joint*

*entropy and conditional entropy) for system analysis and model performance evaluation because they offer several desirable properties: They extend seamlessly from uni- to multivariate data, allow direct comparison of predictive uncertainty with and without model simulations, and their boundedness helps to put results into perspective."(L9-12 in the revised version).*

*In the main text (section 2.4), we believe we have sufficiently justified the use of information measures, and explained them.*

**The introduction effectively positions PB and DD models in the hydrological modeling landscape. That said, the discussion of "limited data" could benefit from greater contextual depth. Recent literature on hybrid or semi-physically based models as a response to data scarcity could be cited to further motivate the study.**

*Thank you for this thoughtful comment. We agree that the hybrid models should be mentioned with their potential advantages regarding data scarce cases and we have added recent literature on this topic. However, we did not go into great depth here, since we think that that is another research avenue that would distract from our main focus. (L46-52 in the revised version).*

**The research questions (Q1–Q4) are well articulated. However, Q3 (related to information content) would be more compelling if its practical utility—such as in guiding monitoring network design or data prioritization—were better explained.**

*We agree. In a revised version of the manuscript, we changed the first part of section 2.5.3 to "Concepts from information theory have been used for a broad range of tasks related to data retrieval and system analysis. For example, Foroozand and Weijs (2021) used it for the optimal design of monitoring networks, Neuper and Ehret (2019) used it to identify the most important predictors for quantitative precipitation estimation, Sippel et al. (2016) used it for a data-based dynamical system analysis. Along the same line, we use information concepts here to find out whether we can predict the success of training a model from a prior analysis of the available multivariate input data (precipitation, temperature, evapotranspiration). Specifically, we investigate whether model performance, expressed by conditional entropy of the streamflow given the model simulation, is predictable from the joint entropy of the input data. These two types of entropy give different insights; joint entropy about the overall variability (information content) of the data; conditional entropy about the information content of the model simulation about the observed streamflow (L389-397 in the revised version).*

**The selection of PB models is appropriate and widely accepted. For DD models, while the choices are generally valid, the authors could briefly justify why other promising alternatives (e.g., NARX networks, Random Forests) were not included.**

*Good point. Currently, LSTMs are the de-facto standard for data-based simulation of hydrological systems, because of their flexible architecture, flexible training and internal memory. LSTMs therefore were an obvious candidate on the list. The other models (EDDIS, RTREE, ANN) are all really thought as lower benchmarks only that reveal how well a simple data-based model can do, if memory is not inherently enabled by the model. We have mentioned this for the EDDIS model in lines 221-234, and for the RTREE model in lines 243-245. We agree that NARX networks and Random Forests should do much better than EDDIS and RTREE. But as stated above, our goal here is to present the range of data-based models, from very simple to the state of the art, which we think is the case with the 4 models we chose.*

*For clarification, in a revised version of the manuscript we have added a sentence at the beginning of section 2.3.2. (L198-203):*

*"We selected four data-driven models with the aim of covering a wide range of model complexity, from very simple ones (EDDIS and RTREE) that serve as a lower benchmark, to simple approaches based on neural networks (ANN) and to the current state of the art (LSTM) that serves as an upper benchmark. Details of each model are described below. Many other data-based methods have been used for modelling hydrological systems, e.g. NARX networks (Renteria-Mena et al., 2023) or Random Forests (Schoppa et al., 2020). These typically show performances between our lower and upper benchmarks, therefore we did not include them in the study for the sake of brevity."*

**The experiment on sampling strategies (random, consecutive, Douglas-Peucker) is a strong point. However, using only the HBV model for this analysis (E2) limits the generalizability of results. Extending this comparison to at least one DD and one additional PB model would add robustness.**

*We are aware that our experiment on sampling strategies included only the HBV model. As emphasised throughout the manuscript and reflected in its structure, this is consistent with our central research objective: to investigate learning behaviour across different model types. While we find the secondary question interesting, a comprehensive investigation would require the inclusion of a broader set of models. Such an extension, while valuable, would shift the scope away from our central research objective. We therefore deliberately refrain from extending the simulations and analysis here, and instead suggest this as a promising direction for future work.*

**The explanation of entropy-based evaluation is informative, though potentially dense for some readers. Including a simplified table comparing conditional entropy with more familiar metrics like NSE or KGE (perhaps in the appendix) would be helpful.**

*Thank you for the suggestion. We included the table below in the Appendix of the revised manuscript, and referred to it in section 2.4. (L327 of the revised version).*

*Table A1: Properties of information-based compared to value-based distance measures between model simulations and corresponding observations.*

| Characteristic | Information measures | Value-based measures |
|---|---|---|
| Examples | Conditional Entropy (CE)

Kullback-Leibler Divergence (KLD) | Mean Squared Error (MSE)

Nash-Sutcliffe Efficiency (NSE)

Kling-Gupta Efficiency (KGE) |
| Distance calculated on | The probabilities of the data values | The data values |
| Distance measured in units of | Bits (if logs are calculated on base 2) | MSE: Squared units of the data

NSE, KGE: [-] |
| Extension to multivariate cases | Straightforward | For NSE and KGE: straightforward |

| | | For MSE: Requires (subjective) choice of weights for the different variates |
|---|---|---|
| Existence of bounds | CE: [0, Unconditional Entropy]

KLD: [0,Inf] | MSE: [0,Inf]

NSE: [-Inf, 1]

KGE: [ -Inf, 1] |
| Mainly sensitive to offsets of | the most frequent events | Large values far from the mean |
| Can be applied to data types | Categorical, numerical | numerical |

**The core finding—PB models plateau early, while LSTM continues to improve with more data—is well demonstrated. The authors could enhance this discussion by elaborating on why LSTM is particularly effective (e.g., its ability to capture long-term temporal dependencies via memory cells).**

*We tried to better explain the outstanding learning of the LSTM by stating: "The LSTM model may outperform the other data-driven models due to its ability to flexibly capture short and long term dependencies, which are essential for modelling hydrological processes." (L463-465).*

**The observed advantage of HBV under random sampling is noteworthy. However, the paper would benefit from discussing whether similar trends are seen in other PB models, such as SWAT+ or mHM.**

*Yes, we agree that it would be interesting to investigate whether that holds also for other models. However, as we mentioned in the response to your comment regarding the sampling strategy experiment, in order to properly address this, we believe that would need to be done for a larger set of models, which is both beyond the scope of the study and takes away from the main focus of the study, which is about the difference in learning behaviour of data-based and process-based models.*

**The discussion is candid and thoughtful, particularly regarding limitations of model performance (e.g., mHM) and sampling methods. One area for improvement is the generalizability of the findings. The study is focused on humid temperate catchments— how well might these conclusions hold in arid, snow-dominated, or tropical regions?**

*Thank you for your positive words! It is indeed valid to ask about the transferability to other regions. We can only speculate here based on what we know from studies of model performance moving from humid to semi-arid and arid regions, where we find generally a decrease in model performance. However, how this is reflected in learnability remains speculative and would certainly be worth addressing in future research. We have added a paragraph on this in the discussion (L656-663): "The choice of our study catchments, all in a humid environment, makes it difficult to draw more general conclusions about the transferability of our results. While model performance tends to decrease when moving from humid to semi-arid or arid regions, we can only speculate about the effects on the learnability of the different models in other regions of the world. For example, in a semi-arid or arid environment, the process-based models may lose some of their advantage in the early stages of learning, as the data pool available for calibration of the storage changes in*

*representativeness. The extent to which the learning curves of the different model types would simply follow a consistent decline in performance from the beginning to the end of the learning curve, or whether this would actually result in different slopes of the curves, is an interesting question for future research."*

**The finding that fully random sampling outperforms the Douglas-Peucker method is intriguing and counterintuitive. The authors should explore potential reasons, such as whether event-based sampling might inadvertently introduce overfitting or neglect broader variability.**

*Like you we were surprised when comparing results of the random sampling with those of the D-P sampling. A potential explanation for the poorer-than-expected results of the D-P method could be that hydrological catchment response is a function of the interplay of short-, intermediate- and long-term storages, which requires adequate parameterization of the related storage functions in the model. Using random sampling selects a time-proportional share of low flow, intermediate flow and high flow situations, which gives the model the opportunity to learn correct parameterizations of baseflow, interflow and fast runoff processes. Sampling by D-P selects the main "turning points" in a time series, which occur at the onset and peak of high-flow events, and thus may leave the model little opportunity to learn about long-term processes. We hypothesize that a hybrid combination of randomly selected points with D-P selected points might yield the best results. We have mention this in the revised version of the manuscript, and leave this for future work. (L722-734 in the revised version).*

**The conclusions are concise and well aligned with the results. However, the broader implications—especially for practitioners—could be emphasized more clearly. For example, LSTM's scalability and adaptability make it promising for ungauged or data-poor basins, and the observed model behavior supports the development of hybrid PB-DD approaches.**

*We follow your suggestion by adding the following in the conclusions: "The LSTM's ability to learn through its flexible approach, combined with the fixed structural architecture that gives process-based models an advantage in data-poor settings, raises the compelling - though as yet unresolved - question of whether hybrid architectures could effectively integrate these complementary strengths."(L719-721 in the revised version).*

**Minor Comment**

**Section 2.3 offers detailed descriptions of model architectures, which could be streamlined. A reference to supplementary material (if available) or a summary table might improve readability without sacrificing depth.**

*We tried to streamline the descriptions of model architectures to make them easier to read and compare. We now also explicitly refer here to the ranges of the parameters of the different process-based models in the supplementary material.*

**Overall Recommendation:**

**This manuscript presents a valuable and methodologically rigorous contribution to hydrological modeling. With minor revisions to better emphasize novelty, expand on**

**generalizability, and clarify certain methodological choices, the paper will significantly enhance understanding of model behavior in data-scarce contexts.**

*Thank you again for your effort, thoughtful comments and the overall positive evaluation. We hope with the above answers we could sufficiently address your comments.*

**Reviewer 2 (Claudia Brauer)**

**Summary**

**This manuscript presents a nice study comparing the learning ability of eight models (four process-based and four data-driven) given different amounts of training data (experiment 1). For subsets of these models, the authors tested if the learning ability is affected by how those training data points are sampled (exp. 2), including previous time steps (exp. 3), and including spatially varying forcing (exp. 4).**

**Overall impression**

**I think the scientific significance, scientific quality and presentation quality are all good. I do have some questions and comments (see below), which hopefully help the authors to improve the manuscript such that readers can better understand the exact methods and interpretation of the results.**

*Thank you for this positive feedback!*

**General comments**

**The title is "a comparison of process- and data-driven models". Only research question / experiment 1 makes a direct comparison between the two types. The other three questions focus on either one type or the other. In addition, in the paper you use the term process-based instead of process-driven, so I would use that in the title also. Maybe "How well do process-based and data-driven hydrological models learn from limited discharge data?" Alternatively, you can distinguish more between the main question (experiment 1) and additional analyses (exp. 2-4).**

*Your title suggestion is much better than the one we had, we took that up.*

**How does the daily resolution compare to the response time of the catchments (especially the smaller subbasins)? Is it enough to capture the dynamics? Please discuss this choice in the text.**

*As we only have daily data available, we cannot make a robust evaluation of how well the response times for fast events are represented. As for other limitations of the study, such as the evaluation of learning only by the simulation performance of discharge and not by other variables, the resolution of the data limits the evaluation for faster process representation. We included that in the discussion by adding "In this study, we have relied on daily data only and have therefore not been able to include an assessment of faster processes on a sub-daily scale. Particularly when using models for specific purposes such as flood forecasting, where these faster processes are relevant, it would be important to include higher resolution data."*

*in Section "3.5 Limitations of the study design" (L632-643 revised manuscript)*

**Discharge time series of one example year for all three catchments would help to get a better feeling for example the response dynamics, the importance of baseflow or snowmelt. This can be positioned in the main text or in supplementary material.**

*Yes, we agree that plotting these time series may be helpful. We have added these in the supplementary material (together with simulated hydrographs from the various models, see also answer to your comment about including KGE values below).*

**Section 2.5: When I first read this part, I thought I had read over information somewhere. In L 351 you say "full length of the time series" - the maximum number 3654 indicates that you use 10 years (out of the 15 available years, as specified in the data section) for training, but I couldn't find where you specified this (and if you always used the same period or tried different options for the training/testing period). I later found this info at the end of the section. Maybe it helps to start Section 2.5 with the last paragraph (modified).**

*Thank you for this suggestion, we moved the last paragraph to the beginning of this section.*

**Did the training or testing period contain extremely high or low flows or other unusual events? Please mention this in Section 2.5.**

*No, both training and testing periods are fairly similar for our time window and catchments. We'll add that information as "Both training and test data periods were very similar in terms of the distribution of high and low flows." (L355)*

**In research question 1 (L 45, Table 4, section header 2.5.1 - check the whole document for other occurrences), you use the word "break point" without making explicit what you mean with this. A break in what? Model performance as a function of sample size? Maybe phrase it something like "at which sample size do data-driven models outperform process-based models"?**

*Thanks for pointing that out. We meant to say that above a certain sample size provided we see different learning behaviour between the different model types. It is indeed not clear how we phrased it in the manuscript and we clarified that by rephrasing it from OLD: ..."break point between process-based and data-driven models?" to NEW: ..."data set size beyond which data-driven models outperform process-based models?"*

**Research question 3 is formulated too complex, which hampers understandability (for me). It reads "Is there a relationship between the information contained in the data and the shape of the learning curve for different models that allows predicting the achievable model performance?", but what you do is assess the effect of including data from previous time steps.**

*We tried to sharpen this research question to "Can we predict learning behaviour of models from an a priori system variability analysis?".*

**In Fig 3 the band is the ensemble of 30 repetitions. If this band expresses the sampling uncertainty, I would expect that the band width becomes zero for large sample sizes because when you use nearly all data points, there is not much choice left for which points you pick. Is the remaining variation really a measure of the sampling uncertainty or actually the calibration uncertainty? I think it's good to point it out in the text and discuss in the discussion section. You do mention it briefly in L 469-470, but I would move this to the first paragraph of the section, so it's easier for the reader to understand the figure.**

*What is shown in Fig. 3 is the evaluation of the simulation results in the validation period from the 30 calibrations using 30 sampling repetitions. These simulations include both the effect of the different replicate samples, which converge as the sample size increases, as you rightly mentioned, and they also include the calibration uncertainty. We have added "The band around the median learning curve indicates both the effect of different replicant samples at each sample size, as well as the calibration uncertainty." L427/428 to make this clear.*

**I understand why you use the entropy as measures (and I don't object to it), but for me it's difficult to interpret if the models are all very good to start with and improved somewhat, or that they were rubbish to start with and became slightly better. Putting the KGE plots in the supplementary is fine, but I would include some KGE values in the text. I would appreciate some time series of the simulations (in the supplementary material and referring to it in the main text is fine). It may also be to split the KGE into components or use separate metrics to assess the simulation quality. For example, I can imagine that a bias in the forcing will be much more harmful to a process-based (water balance conserved) model than to a data-driven model.**

*We also had to fight internally to keep the more classical performance metrics only in the supplementary material. One point we wanted to make in our paper is how we can use the entropy usefully in hydrological modelling. We appreciate your thought about using the components rather than the full KGE to see differences, think however that this is beyond the scope of the study, since we are not evaluation each process closely but in an average learning behaviour.*
*The dashed line in Fig 3 gives the maximum entropy, which can be seen as low benchmark assuming uniform predictive distribution. We highlighted the function of this line by adding the following text in the captions: "The dashed line shows the maximum possible entropy, which can be used as a benchmark, in a similar way to how the mean discharge prediction is used in the Nash-Sutcliffe-efficiency.".*
*We have also added more information in the form of plots of the simulated hydrographs in the supplementary material, as any classical performance metric also hides/highlights different aspects.*

**Could you zoom in to a few specific periods in the testing period that has high relevance for water managers (flood, drought, snowmelt peak) and see how the simulation of those are affected by sample size and model choice? Simply showing the time series with medians (with or without uncertainty bands) probably suffices to illustrate the practical usefulness.**

*While we appreciate the idea, we believe that this detailed examination of specific processes is beyond the scope of this very study, which, in our opinion, already covers various aspects. We would stick to the information metrics and the additional hydrographs in the supplement, potentially using that idea potentially in a future study.*

**Figure 3 is in my opinion the most important result and I think it deserves some more space such that it's easier to distinguish the curves, especially at low sample sizes. Some suggestions: (1) Put the graphs below each other, with a shared x-axis. Putting the panels above each other makes it easier to compare at which sample size the learning curve flattens. I agree that it's not necessary to make the y-axes the same in these plots (also for figs 5 and 6). (2) I think (but please try to see if it works out well) that it would help to make the x-axis logarithmic to make it easier to see what's happening at small sample sizes. Your choices for sample sizes (2, 10, 50, 100, 250, 500, 1000, 2000, 3000 and 3654) would also justify a logarithmic x-axis. (1) Reduce the font size to match the size of the main text (do this also for the other figures). (3) Make the panels wider, such that they fill a column in the final paper, and a bit higher. You can do the same for fig 5 and 6. Other than that, the figures are effectively designed.**

*Thank you for the specific and helpful suggestions. We implemented them all except the logarithmic scale. We found that it hid the most interesting saturation points, so we did not apply it.*

**In Section 2.5 you explain the sampling strategies. It was not clear to me if the samples with different sizes are independent or not. In other words, do you first pick 2 data points and then add 8 data points to it for the 10-data-point-sample, or do you pick 10 new points? This matters for the interpretation of Figure 5. In L 508 you mention "repetition lines", so I expect you used the same points and added to it, else they would not be lines but separate points (which would also be fine, but then you cannot connect them with a line).**

*The samples are independent from each other. Lines are there only for visualization and we clarified this in the caption.*

**On which models was experiment 3 performed? I suppose EDDIS, RTREE and ANN? Mention this again at the start of Section 3.3.**

*Thank you for raising this question, it reminded us to be more specific at this point. All joint and conditional entropy values calculated for experiment 3 were calculated without a model, but simply based on the available data. This is equivalent to the EDDIS "model". In the revised version of the manuscript, we will mention this at the beginning of section 3.3. L449-551*

**L 556-557, L 674-677, L 704-706 (maybe other locations as well): You say that high conditional entropy indicates high variability. Can you explain this a bit more? What type of variability do you mean exactly? Simply the variance? Is the high variability mostly caused by large seasonal dynamics or flashy response to rain events or snowmelt? This may affect the results, since a memory of 7 days is not long enough to capture seasonal dynamics. Can you relate this variability to the discharge dynamics found in the catchments, for example as shown in a time series plot or expressed in other metrics (for example flow duration curve slope or baseflow index)?**

*Thank you for raising this question, like the previous one it reminded us to be more specific about what we mean by "variability". By variability we mean the spread of the empirical multivariate distribution of the data, and we express it by joint entropy Hj. This is different from variance in the sense that is calculated from the probability distribution of the data rather than the data themselves. We calculated Hj including memory effect by adding time-*

*aggregations of the input data, but for the sake of clarity only for a few selected cases of (see tables 6 and 7), and with a maximum memory depth of 6 days. We did so because typically, streamflow at a timestep t is usually mainly influenced by rainfall of the past few days. Longer-term seasonal variations are therefor not included, and would be beyond the scope of our study. The relation of the data variability (table 6) to predictability is established by calculating the conditional entropy of the streamflow given the input data (table 7). Please see the related discussion in section 4. In the revised version of the manuscript, we have added further clarifications in sections 3.3, 3.6 and 4.*

**L 658-659 "the Selke catchment has a very distinct sub-basin with a steep topography, which is not present in all lowland catchments". I (but I am biased) would indeed not classify the Selke basin as a lowland, with nearly 500 meters of elevation difference. I think it's fine that you use this catchment, but you cannot use it as a representative for lowland catchments in general or the central German lowlands in particular. You can make it more clear in this sentence and also in L 7, L 89, L 699, L 724 (for example replace "in the central German lowlands' with "on the transition between the Harz mountains and the central German lowlands" or "at the edge of the central German lowlands"). Check the whole document for references to this catchment as lowland catchment.**

*Agreed. We checked carefully and changed where it was mentioned as a lowland catchment or a representative thereof.*

**In the conclusion you switch the order of the experiments (1-2-4-3). I think this order makes sense since experiments 2 and 4 have similarities in terms of the model used. I would advise to use this order throughout the paper.**

*We chose the order of the research questions according to their priority for the main text. You are right that the order in the conclusions is not consistent with the order of rest of the study, however this appeared meaningful to avoid repetitions there and keep it concise.*

**Specific comments**

**Abstract is good, but quite long. Can you make it more concise?**

*We tried to do that as much as feasible and propose the following revised version:*
*"It is widely assumed that data-driven models only achieve good results with sufficiently large training data, while process-based models are usually expected to be superior in data-poor situations. To investigate this, we calibrated several process-based and data-driven hydrological models using training datasets of observed discharge that differed in terms of both the number of data points and the type of data selection, allowing us to make a systematic comparison of the learning behaviour of the different model types. Four data-driven models (conditional probability distributions, regression trees, ANN, and LSTM) and three process-based models (GR4J, HBV and SWAT+) were included in the testing applied in three meso-scale catchments representing different landscapes in Germany: the Iller in the Alpine region, the Saale in the low mountain ranges, and the Selke in the transition between the Harz and Central German lowlands. We used information measures (joint entropy and conditional entropy) for system analysis and model performance evaluation because they offer several desirable properties: They extend seamlessly from uni- to multivariate data, allow direct comparison of predictive uncertainty with and without model simulations, and their boundedness helps to put results into perspective. In addition to the main question of this*

*study — to what extent does the performance of different models depend on the training dataset? — we investigated whether the selection of training data (random, according to information content, contiguous time periods or independent time points) plays a role. We also examined whether the shape of the learning curve for different models can be used to predict the achievable model performance based on the information contained in the data, and whether using more spatially distributed model inputs improves model performance compared to using spatially lumped inputs. Process-based models outperformed data-driven ones for small amounts of training data due to their predefined structure. However, as the amount of training data increases, the learning curve of process-based models quickly saturates and data-driven models become more effective. In particular, the LSTM outperforms all process-based models when trained with more than 2-5 years of data and continues to learn from additional training data without approaching saturation. Surprisingly, fully random sampling of training data points for the HBV model led to better learning results than consecutive random sampling or optimal sampling in terms of information content. Analyzing multivariate catchment data allows predictions about how these data can be used to predict discharge. When no memory was considered, the conditional entropy was high. However, as soon as memory was introduced in the form of the previous day or week, the conditional entropy decreased, suggesting that memory is an important component of the data and that capturing it improves model performance. This was particularly evident in the catchments the low mountain ranges and the Alpine region."*

**137-139 "Gridded values for temperature and precipitation were obtained by interpolating the observations from meteorological stations.": Can you give an indication of how many stations for precipitation and temperature are inside the catchments, or the average density? To get an idea of the spatial coverage?**

*According to the DWD, the base station data for gridded interpolated data changes over the years. For the period under consideration in our study, there are approximately 20 stations in the Selke catchment, 30 stations in the Saale catchment, and 30 stations in the Iller catchment. However, we could not find the exact number of stations that went into the interpolation algorithm. We included this approximate number of stations in the revised version of the manuscript. L143/144*

**Table 1: Add mean annual potential ET to the table. You could also add actual ET (computed by P-Q). This gives some idea of the climatology and relevant processes. Instead of nival/pluvial, can you specify which percentage of the precipitation falls when the temperature is below zero?**

*Yes, we included ET as suggested but keep the regimes, since the snow estimate would be from a simple snow model rather than long term observations of effects in discharge.*

**Table 2: Add references for the DWD datasets for precipitation and temperature. Also for the soil map.**

*We have added these.*

**139: Why did you choose Hargreaves and not a more accurate method involving more meteorological variables (e.g. Penman-Monteith)?**

*We had to rely on data that was provided in the research network and did not focus on other*

*approaches here. We also think that for the learning that should not make a large difference at least regarding discharge. If the study would be broadened or the focus changed to other hydrological processes it may become important.*

**L 192 " Therefore, mHM often provides good results without calibration, using the default parameter set." Did you do that as well or did you calibrate some parameters. Please specify how many parameters you calibrated.**

*No, the default parameter set was not used at all in the study. The number of calibrated parameters could be seen in the parameter table in the supplement. However, we plan to remove the mHM results from the study entirely (see last comment below).*

**L 204" large number of parameters": how many did you calibrate?**

*The calibrated parameters are in the supplement. We added a reference to the respective table in the main text.*

**Did you use any normalization for the data-driven models?**

*For EDDIS, the data were categorized into bins covering the data range without prior normalization. For RTREE, data were used without prior normalization. For ANN and LSTM, data were standardized before use, i.e. subtraction of the mean and division by the standard deviation.*

**Table 3: it would help to add the time steps used in EDDIS, RTREE and ANN to P, T and PET.**

*All components that are comparable among models are shown in this table and we believe adding these time steps that are not common for all other models would make the table less clear. However, we tried to make this very clear revised text.*

**Move Table 5 up in the manuscript (you refer to it already in line 206). I would actually merge tables 3 and 5.**

*We have merged the tables.*

**Check if you explained all symbols in the equations. Eq. 1: mention in the text what x is (x1,xn) and what j (in Hj) stands for (I suppose joint). Eq. 2: mention Hc, Y, y. Eq. 3: Lrel.**

*Thank you for pointing us at this. In the revised version of the manuscript, we added an explanation of*

*- x1, x2 (realisation of the random variables X1, X2,...)*

*- Hj (the "j " here indeed indicates "joint", to distinguish it from Hc, where the "c" is for conditional*

*- y as a realisation of the random observed target variable Y*

*- Lrel is the measure of relative learning*

**Table 4 should include the data-driven models as well.**

*Table 4 includes also the data driven models = all.*

**L546 "A major advantage of random sampling over optimal sampling is the ability to repeat the sampling". I expect that it's also a disadvantage that you don't know on forehand when the optimal moments will occur. In addition, the measurement uncertainty is often highest during peak discharges.**

*Yes true, but for the optimal samples we don't know either beforehand, only if we sample afterwards to reduce the sampling size to be used and if we would do event sampling.*

**Merge tables 6 and 7 (then it's easier to compare the numbers).**

*The values in the tables are not directly comparable and only the values within each table should be compared. Keeping the tables separate may therefore reduce the temptation to directly compare values from one table to another. We added a sentence in each of the captions stating that this is the case and that direct comparison between tables should be avoided.*

**L 724 "the lowland catchment, which showed the least improvement". Is this correct? Wasn't the Saale catchment the one with the least improvement?**

*No, actually the least improvement could be seen for the Selke catchment.*

**Supplement: Placement of figures and tables is confusing. Maybe adding hard page breaks helps to structure it. First present the tables with parameter values and then the figures with results. Check if you refer in the main text to each of the tables and figures in the supplement.**

*Thank you we adjusted them as suggested.*

**Textual comments**

**L 44: Do you mean validation or variation? If the first, please explain.**

*Yes, variation. We changed it in the revised version to variation.*

**Fig 1, caption: location -> locations**

*Changed.*

**Table 3 caption and L 554: pET -> PET**
*Changed.*

**Table 3: remove all the "daily", since they are the same everywhere. You could add it in the caption.**

*Removed.*

**L 273: To set up the process-based models, some of them make use of additional data -> Some of the process-based models require additional data for the set-up**

*Changed.*

**L 297 few -> little (or: a small amount of)**

*Changed.*

**L 330: (Gupta et al. (2009)) -> (Gupta et al, 2009)**

*Changed.*

**L 350: ten different sample sizes -> ten sample sizes (different is a redundant word)**

*Changed.*

**L 380: for to-> to**

*Changed.*

**L 383: changes for -> varies between (if I understand you correctly)**

*Changed.*

**L 387: three different sampling schemes -> three sampling schemes**

*Changed.*

**L 428: that all models decrease the conditional entropy -> that for all models the conditional entropy decreases with increasing sample size**

*Changed.*

**L 445: it remains in the performance-> its performance remains**

*Changed.*

**L 446 data based -> data-driven (check the whole document for consistency with process-based and data-driven)**
*Changed.*

**Fig 4: Swap the order of the bars, to match the order you use in the whole document and legend (Iller on top)**

*Adapted.*

**L 592: Iller ->Iller and Selke**

*Changed.*

**L 690: done: While-> done. While**

*Changed.*

**L 708: is changing -> changes / varies**

*Changed.*

**Fig S1: too small to see anything. Zoom in to one year and spread it out over the full width, or show all years spread out over multiple lines.**

*Adapted.*

**Fig S4, y-label: Hc -> KGE**

*Changed.*

**Fig S4 caption: Three different sampling schemes -> Lumped/distributed forcing**

*Changed.*

---

## Author Response (AR2)

Dear Nunzio Romano,

We are delighted to receive this positive feedback. We would like to express our appreciation once again for the careful and constructive reviews provided by both reviewers, and thank you for your efforts with our manuscript.

Sincerely, also on behalf of all the co-authors.

Maria Staudinger